EMBO
Molecular Medicine

# LXR pathway drives hormonal response intensity in polycystic ovary syndrome

Sarah Dallel[1,2,3,4], Manon Despalles[1,2,3,4], Margaux Tore[1,2,3,4], Yoan Renaud [1,2,3,4], Ayhan Kocer[1,2,3,4], Christelle Damon-Soubeyrand[1,2,3,4], Pierre Pouchin[1,2,3,4], Caroline Vachias [1,2,3,4], Katia Boutourlinsky[1,2,3,4], Céline Gonthier-Gueret [1,2,3,4], Angélique De Haze[1,2,3,4], Phelipe Sanchez[1,2,3,4], Jean-Christophe Pointud [1,2], Erwan Bouchareb[1,2,3,4], Marine Vialat [1,2,3,4], Aurélie Lagarde[1,2,3,4], Cristina Gulunga[5], Laure Chaput [5], Aurélie Vega [5], Florence Brugnon [3,5,6], Igor Tauveron[4], Amalia Trousson [1,2,3,4], Cyrille de Joussineau [1,2,3,4], Françoise Degoul [1,2,3,4], Laurent Morel[1,2,3,4], Jean Marc Lobaccaro [1,2,3,4], Salwan Maqdasy [1,2,3,4,7] & Silvère Baron [1,2,3,4✉]

## Abstract

**Gonadotropin injections used to stimulate oocyte production during assisted reproductive technology (ART) procedures are associated with the risk of an abnormal response in predisposed patients suffering polycystic ovary syndrome (PCOS). Liver X receptors (LXR) pathway has been identified as key regulators during this process. This study explores the integration of the hormonal signals, cellular networks and molecular mechanisms linking sterol signaling with inflammation and immune infiltration. Pharmacological activation of LXR in a wild-type context protects against gonadotropin hyperstimulation mirroring the effect observed in LXR-deficient mice. Ovarian stimulation leads to immune cell infiltration orchestrated by granulosa cells in absence of LXR, resulting in an altered granulosa cell response to gonadotropin and enhanced inflammation. LXR controls inflammasome activity by regulating Thioredoxin Interacting Protein (TXNIP) gene expression in mural granulosa cells, thereby modulating IL1β production. This immune cell infiltration persists throughout ovulation in PCOS patients and is observed in cumulus oocytes complexes, highlighting the pivotal role of LXR path in regulating inflammatory processes during hormonal stimulation in ART procedures.**

**Keywords** Polycystic Ovary Syndrome; Granulosa Cells; Liver X Receptors; Hormonal Stimulation; Inflammasome
**Subject Categories** Immunology; Metabolism; Urogenital System

## Introduction

The administration of gonadotropins for assisted reproductive technology (ART) procedures, akin to the contraceptive pill, represents a paradox in medicine, given the prescription to healthy patients with no curative purpose. In patient with PCOS, ovulation induction is a classical option to circumvent infertility. Surprisingly, there is no gold-standard treatment for ovarian stimulation, which leads to a great heterogeneity in the medical patient care (Lunenfeld, 2012). A risk of complication following treatment with gonadotropins in polycystic ovary syndrome (PCOS) patients is an ovarian hyperstimulation syndrome (OHSS). Although some molecular mechanisms have been proposed, such as the high production of Vascular endothelial growth factor (VEGF) (D'Ambrogio et al, 1999; Doldi et al, 1999) in response to treatment, the molecular etiology of this hyper-response is still debated. In 2009, Mouzat et al, described an OHSS-like phenotype in Liver X Receptors (LXRs) double knock out (DKO) mice invalidated for both LXRalpha and LXRbeta. They identified for the first time a transcription factor located upstream of the effectors already known, enabling a better understanding of the altered molecular pathways at the origin of this abnormal ovarian response (Mouzat et al, 2009).

Liver X receptors belong to the nuclear receptor superfamily and are transcription factors controlling the expression of genes involved in carbohydrate and lipid metabolism as well as in immunity (Dallel et al, 2018; Hong and Tontonoz, 2014). Like peroxisome proliferator-activated receptors γ, α, and δ receptors (PPAR), LXRs have been reported to inhibit pro-inflammatory gene expression (Steffensen et al, 2013; Joseph et al, 2003). This function is particularly linked to the existing crosstalk between LXRs and the Nuclear Factor-kappa B (NF-κB) signaling pathway (Joseph et al, 2003; Castrillo et al, 2003). Furthermore, several studies have shown that the majority of the genes responsible for

[1]Université Clermont Auvergne, iGReD, CNRS UMR 6293, INSERM U1103, 28, place Henri Dunant, BP38, 63001 Clermont-Ferrand, France. [2]Groupe Cancer Clermont Auvergne, 28, place Henri Dunant, BP38, 63001 Clermont-Ferrand, France. [3]Centre de Recherche en Nutrition Humaine d'Auvergne, 58 Boulevard Montalembert, F-63009 Clermont-Ferrand, France. [4]Service d'Endocrinologie, Diabétologie et Maladies Métaboliques, CHU Clermont Ferrand, Hôpital Gabriel Montpied, F-63003 Clermont-Ferrand, France. [5]Assistance Médicale à la Procréation, CECOS, CHU Clermont-Ferrand, F-63003 Clermont-Ferrand, France. [6]Université Clermont Auvergne, IMOST, INSERM 1240, Faculté médecine, 28, place Henri Dunant, BP38, 63001 Clermont-Ferrand, France. [7]Present address: Department of Medicine (H7), Karolinska Institutet, Karolinska University Hospital Huddinge, Huddinge, Sweden. ✉E-mail: silvere.baron@uca.fr

producing pro-inflammatory mediators, e.g., those under the control of the transcription factors Activator Protein-1 (AP1) and NF-κB, can be repressed by LXR in the absence of LXRE-binding sites (LXRE) within their promoter sequences (Pascual and Glass, 2006). Thus, LXR exerts this control of pro-inflammatory pathways mainly through trans-repressive mechanisms (Ogawa et al, 2005).

Inflammation is a physiological process mobilized during pathological development. Inflammation is therefore a mechanism under an orchestrated control making possible to adjust the amplitude of the response by integrating various extrinsic or intrinsic stimuli. One of the mechanisms allowing this control is the inflammasome (Lu et al, 2014; Rathinam and Fitzgerald, 2016). This consists in an oligomerization complex of several proteins, NOD-like receptor family, pyrin domain containing 3 (NLRP3), Apoptosis-associated speck-like protein containing a CARD (ASC) and Caspase-1 (CASP1), building a complex able to carry out the proteolytic cleavage of two pro-cytokines, pro-IL1β and pro-IL18 into mature inflammatory mediators (Kayagaki et al, 2015; Johnston et al, 2005). This cleavage is essential to obtain active IL1β and IL18 as well as to be secreted and ensured pro-inflammatory functions. Notably, the role of the inflammasome in the ovary has been identified by analyzing the phenotype of Nlrp3-/- mice (Lliberos et al, 2020; Navarro-Pando et al, 2021). Ablation of NLRP3 results in an increase of age-related fertility and delays the ovarian aging. These observations showed that inflammasome function is closely related to ovarian physiology and its over-activity could be related to ovarian dysfunctions.

We then explored molecular mechanisms downstream of LXR path that could be involved in altered gonadotropins responses and drawn perspectives regarding PCOS patients.

# Results

## Pharmacological targeting of LXR protects against gonadotrophins hyperstimulation

We had previously observed that LXR DKO mice exhibit an OHSS-like phenotype in response to hormonal stimulation, suggesting that LXR have a protective against hyperstimulation (Mouzat et al, 2009). To explore this potential effect, we exposed wild type mice to either a standard stimulation protocol, hyperstimulation protocol only, or hyperstimulation protocol with additional LXR co-stimulation (Fig. 1A). The hyperstimulation protocol has already been described to mimic exaggerated response hallmarks (Chuderland et al, 2013), and involves two consecutive injections of pregnant mare serum gonadotropin (PMSG) and a single high dose of human chorionic gonadotropin (hCG) compared to classical stimulation. To maintain continuous LXR stimulation, four GW3965 injections starting 24 h before the first PMSG treatment and repeated every 24 h were applied. After 96 h, ovarian phenotype was evaluated by ovarian weight index (Appendix Fig. S1) and histological HE-staining. As expected, supra-pharmacological stimulation of wild-type mouse ovaries led to hemorrhagic phenotype (Fig. 1B,C). Notably, GW3965 co-treatment protected against the hyperstimulation phenotype by significantly decreasing the number of hemorrhagic cysts (Fig. 1C). Next, considering that the granulosa cell compartment represents the principal target of PMSG treatment to support folliculogenesis,

we explored the specific role of LXR in these cells. We compared LXR DKO mice, known to exhibit OHSS phenotype under classical hormonal protocol (Mouzat et al, 2009), to TgAMH-Lxrβ rescued mice that re-expressed LXRβ in granulosa cell in a genetic LXR DKO background (Maqdasy et al, 2015). LXRβ granulosa-specific expression fully rescue the hemorrhagic follicles phenotype (Fig. 1D). Granulosa dysfunction in LXR-null ovaries has been confirmed by the downregulation of specific marker expressions such as Cyp19a1, Fshr, and Inha. Interestingly, expression of the latter was fully restored in the TgAMH-Lxrβ rescue model (Fig. 1E). The number of oocytes collected into the oviduct after ovulation following stimulation is characteristic of the OHSS phenotype. We observe a double increase in oocytes produced in DKO LXR animals compared to wild types (Fig. 1F). The number of oocytes produced by TgAMH-Lxrβ is equivalent to wild-type mice. As already reported (Mouzat et al, 2009), oocyte quality evaluation displayed fewer normal oocytes in LXR DKO mice. Again, this normal oocyte ratio was normalized in TgAMH-Lxrβ mice (Appendix Fig. S2). Both observations confirm the central role played by LXRβ in granulosa cell to drive PMSG response. In parallel, OHSS is associated with a vascular phenotype. Thus, we explored the ovarian vascular tree using an endothelial marker, endomucin, and we observed a significant increase in the size of the vascular network both in LXR DKO and TgAMH-Lxrβ mice (Appendix Fig. S3A). This alteration is accompanied by an increase in vascular permeability (Appendix Fig. S3B). As expected, rescue of LXRβ in granulosa cells erases the hemorrhagic and pro-inflammatory phenotype within the follicles but does not correct the vascular defects. Together, these results indicate LXRβ-dependent regulation is necessary in granulosa cells to ensure a proper response to gonadotropin stimulation of the ovary.

## Hormonal ovarian stimulation is associated with immune cell infiltration in the absence of LXR

The development of hemorrhagic cysts was monitored throughout the hormonal stimulation (Fig. 2A). A significant increased number of hemorrhagic cysts was observed early, at 40 h post-PMSG injection, indicating the relevance of this time point for identifying the molecular signature of the phenotype (Fig. 2B). TgAMH-LXRβ expression could rescue hemorrhagic cysts phenotype at all temporal windows tested, confirming the previous observation suggesting to the prominent role of LXRβ in granulosa cells during hormonal stimulation (Fig. 2B). To determine the transcriptomic signature associated with LXR DKO phenotype, we performed RNA sequencing (RNAseq) comparing ovary samples from wild-type, LXR DKO and TgAMH-LXRβ mice. Principal components analysis demonstrated the reproducibility of the biological triplicates (Fig. 2C). As expected, the first dimension efficiently clustered wild-type from LXR DKO and TgAMH-LXRβ samples. Clustering separation between LXR DKO and TgAMH-LXRβ was less pronounced, given that granulosa rescued-cells may represent a minor fraction of the total ovary cell population. Next, using the Venn approach, we identified 258 genes (241 up- and 17 down-regulated) commonly deregulated between wild type/LXR DKO and TgAMH-LXRβ/LXR DKO comparisons (Fig. 2D; Appendix Fig. S4, Dataset EV1). To obtain a gene ontology signature, we then compared independently wild-type and LXR DKO from TgAMH-LXRβ and LXR DKO samples. This strategy led us to identify gene enrichment lists that are closely related between each comparison and pointed towards a marked signature associated

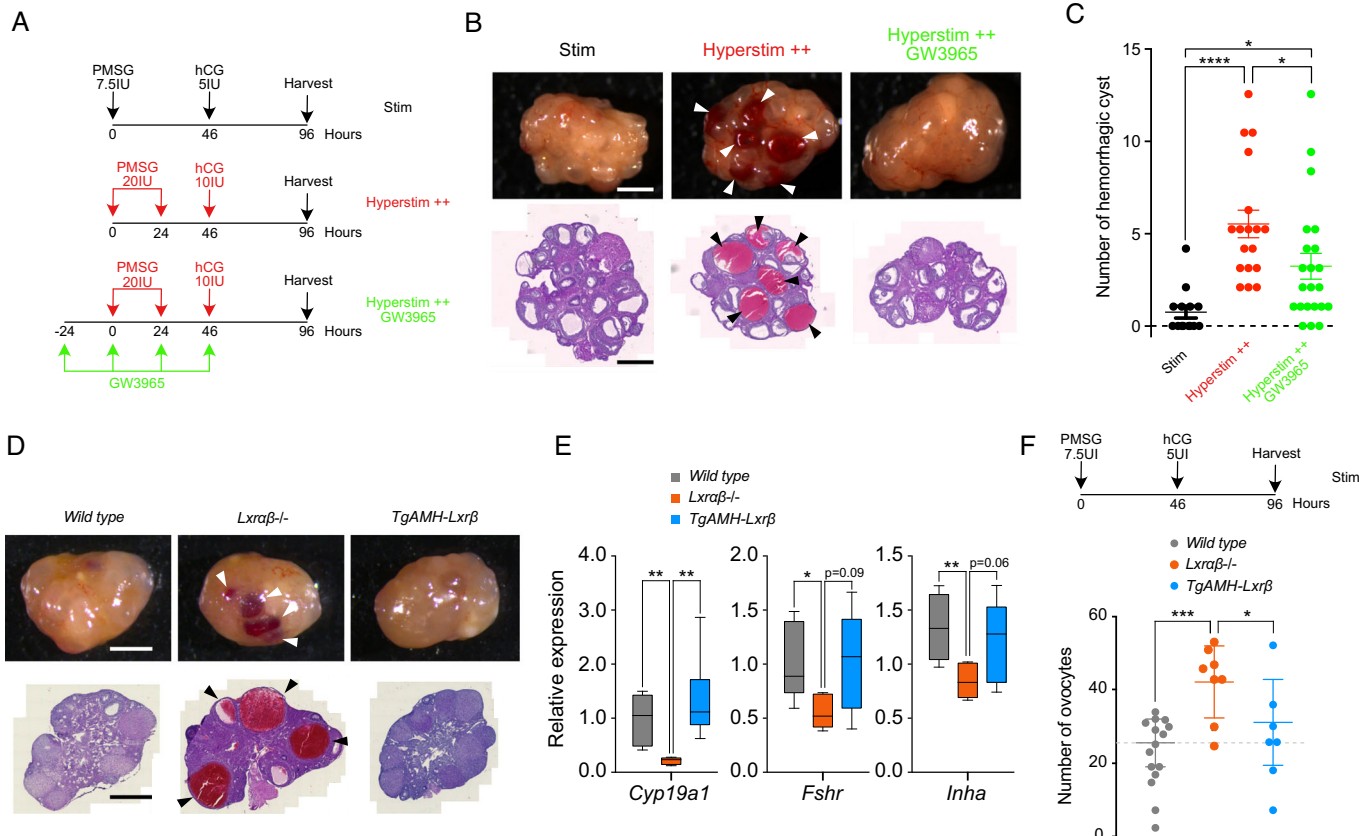

**Figure 1. LXRs activation prevents ovary from gonadotrophins hyperstimulation.**

(**A**) Stimulation protocols: mice receive a single PMSG IP injection (7.5IU) and hCG IP injection (5IU) 46 h later (Stim) $n = 14$, or two IP injection (20IU) at hour 0 and 24 following a single hCG IP injection (10IU) (Hyperstimulation ++) $n = 18$ and a similar protocol with additionnal GW3965 (20 µg/mL) treatments 24 h before starting the protocol and together with following PMSG/hCG IP injection (Hyperstimulation ++ GW3965) $n = 22$. (**B**) Macroscopic representative pictures (upper panel) and HE-staining (bottom panel) of mouse ovaries after 96 post-hormonal protocols. Arrows indicated hemorrhagic cysts (Scale bars = 1 mm). (**C**) Hemorrhagic cysts have been counted in each group. (**D**) Macroscopic representative pictures (upper panel) and HE-staining (bottom panel) of wild type, LXR DKO, and TG-AMH-Lxrβ mouse ovaries following hormonal stimulation. Arrows indicated hemorrhagic cysts (Scale bars = 1 mm). (**E**) Expression of granulosa-specific markers: *Cyp19a1*, *Fshr*, and *Inha* were analyzed by RTqPCR from wild type $n = 6$, LXR DKO $n = 6$, and TG-AMH-Lxrβ $n = 6$ ovaries. Gene expressions were normalized using *36b4* gene expression. (**F**) Total number of oocytes retrieved in oviduct after ovulation following stimulation protocol from wild type $n = 13$, LXR DKO $n = 8$ and TG-AMH-Lxrβ $n = 7$ mice. In (**E**), boxes extend from the 25th to 75th percentile, the middle line shows the median, whiskers extend to the most extreme data. In (**C**) and (**F**), averages values ± SD are represented. Significance determined in (**C**) and (**F**) by Ordinary one-way ANOVA and in (**E**) by Mann and Whitney test. *$P < 0.05$, **$P < 0.01$, ***$P < 0.001$, ****$P < 0.0001$ (exact $P$ values for these statistical comparisons are shown in Appendix Table S1). Source data are available online for this figure.

with immune cells (Fig. 2E). To identify a potential immune cell infiltration, we then performed CD45 immunostaining in LXR DKO ovary and compared it with wild-type and TgAMH-LXRβ. As suggested by RNAseq signature, LXR DKO ovaries specifically displayed a significant increase in the number of immune infiltrated cells visualized by CD45 staining (Fig. 2F,G). To further characterize the nature of the infiltrate, we used CIBERSORTx platform as a tool to estimate the immune cell type abundances. This approach confirmed the increased of the immune infiltration in DKO mice (Fig. 2H) but did not reveal significant changes in the composition of the infiltrating cells in terms of percentage whatever the genotype.

LXRs are knows as key players in immune cell responses (Zelcer and Tontonoz, 2006), prompting us to investigate whether immune response could be the consequence of a potential leak of LXRβ transgene expression under the Anti-müllerian hormone (AMH) promoter in immune compartment. To challenge immune cells, we transplanted bone marrow originated from wild-type, LXR DKO as

well as TgAMH-LXRβ mice into NOD scid gamma (NSG) hosts and performed a hormonal stimulation (Fig. 2I). We first checked the efficiency of bone marrow grafting by analyzing circulating B lymphocytes in blood samples. As expected, NSG mice are depleted in B cells and bone marrow xenograft from each donor restored a normal flow cytometry profile (Fig. 2J). We next stimulated mice with PMSG and hCG as previously performed. CD45 staining showed that wild-type, LXR DKO, or TgAMH-LXRβ grafted NSG mice led to the restored presence of immune cells in the ovary tissue regardless of the bone marrow genotype origin (Fig. 2K). Quantification exhibited that the number of infiltrated cells was identical in all grafted NSG mice indicating that intrinsically, immune cells respond similarly to ovary stimulation independently of their LXR status (Fig. 2L). We confirmed that most infiltrated immune cells derived from bone marrow grafting, as they express the major histocompatibility complex II (MHC-II) marker, which is missing in the NSG background (Fig. 2K). Thus, we could conclude that the increase in immune infiltration observed in

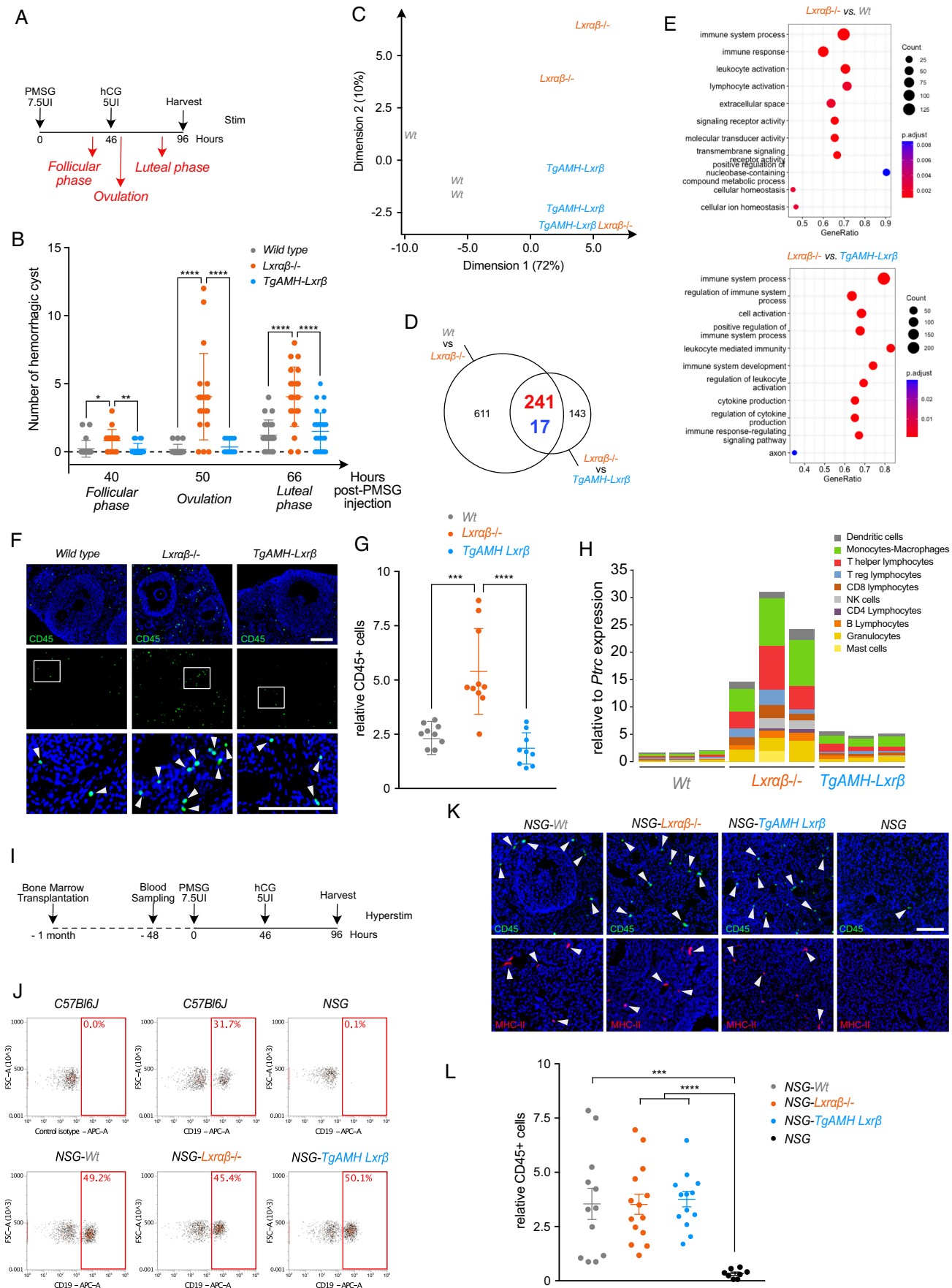

Figure 2.   LXRs ablation is associated with immune cell infiltration within the ovary.

(A) Kinetic chart analysis of phenotype occurrence. (B) Hemorrhagic cysts quantification 40 h, 50 h, and 66 h post-PMSG injection observed in wild type, LXR DKO, and TG-AMH-Lxrβ ovaries (n for each comparison are shown in Appendix Table S1). (C) Principal component analysis of RNAseq dataset 40 h post-PMSG stimulation. (D) Venn diagram comparing wild type versus LXR DKO and LXR DKO versus TG-AMH-Lxrβ. (E) Gene ontology analysis using Cluster Profiler obtained from wild type versus LXR DKO (upper panel) and LXR DKO versus TG-AMH-Lxrβ (bottom panel) comparisons. (F) Immunodetection of CD45 (pan-immune cell marker) in green performed on wild type, LXR DKO, and TG-AMH-Lxrβ ovaries (Scale bars = 100 μm). (G) Quantification of CD45 staining performed on wild type n = 10, LXR DKO n = 10 and TG-AMH-Lxrβ n = 10 ovaries. (H) Analysis of infiltrated cell composition using CIBERSORTx platform conducted on RNAseq data. (I, J) Protocol for bone marrow transplantation following hormonal ovarian stimulation. After 1 month of bone marrow transplant, circulating B lymphocytes were analyzed 48 prior hormonal stimulation using C57BL6J, NSG and transplanted-NSG for each genotype donor. (K, L) Immunodetection of CD45 and quantification staining. C57BL/6 donor-specific MHC-II staining. NSG-Wild type n = 12, NSG-LXR DKO n = 14, NSG-TG-AMH-Lxrβ n = 13 and NSG n = 8 mice have been used. White arrows indicate infiltrated cells (Scale bars = 100 μm). Averages values ± SD are represented. Significance determined in (B) by Ordinary one-way ANOVA, in (G) by Mann and Whitney test and in (L) by Kolmogorov–Smirnov test. *$P < 0.05$, **$P < 0.01$, ***$P < 0.001$, ****$P < 0.0001$ (exact P values for these statistical comparisons are shown in Appendix Table S1). Source data are available online for this figure.

LXR DKO mice was not the consequence of LXR depletion in the immune cells themselves but was, in fact, due to impaired granulosa cell hormonal response.

## Hormonal stimulation is associated with granulosa cell-dependent inflammation due to LXR depletion

To identify the molecular signal initiating immune cell infiltration, we analyzed RNA sequencing data using a dataset obtained from ovaries collected 40 h post-PMSG induction, employing gene set enrichment analysis (GSEA). As in our previous trials, we compared wild-type or TgAMH-LXRβ with LXR DKO mice independently, relying on hallmark gene lists. The sixth-highest enrichment scores for each comparison corresponded to lists of genes linked to inflammation and NF-κB pathway deregulation (Fig. 3A). To explore this proinflammatory NF-κB-dependent signature, we constructed a heatmap using a canonical list of NF-κB target genes (Fig. 3B, Dataset EV2). Surprisingly, all these genes were found to be upregulated in the absence of LXR. These deregulation profiles were confirmed by RT-qPCR expression analysis of Nfkbia, Irf1, Csf1, and Spi1 (Fig. 3C). To investigate a potential deregulation of the NFκB signaling in granulosa cells, we detected p65 whose localization in the cytoplasm or the nucleus indicates pathway activity. While wild-type and TgAMH-LXRβ tissues exhibited the typical honeycomb staining associated to cytoplasmic accumulation, nuclear accumulation of p65 was detected in some follicles of LXR DKO ovaries (Fig. 3D). In addition, comparing mural and cumulus granulosa cells regarding p65 localization we observed that nuclear staining is mostly abundant in mural cells of LXR DKO ovaries (Appendix Fig. S5). Taken together, these observations indicate that the presence of LXRβ expression in the mural granulosa compartment is required to control inflammation during hormonal stimulation of the ovary.

## LXR null mice display a specific granulosa response to hormonal stimulation

To further investigate how granulosa cell response to PMSG treatment initiates immune infiltration when LXR are absent, we decided to explore the specific signature of granulosa cells to PMSG. Among this list of genes, we aimed to identify those whose expression levels are altered by the absence of LXR. First, we analyzed transcriptome profiles of granulosa cell isolated from wild-type mice expose to PMSG treatment for 48 h (Fig. 4A) (Madogwe et al, 2020). We identified genes that

respond to PMSG (Dataset EV3, Appendix Fig. S6). Using this list of genes, we performed GSEA analysis to determine if some of them were deregulated in LXR DKO ovaries 40 h post-PMSG injection compared to both the wild-type and TgAMH-LXRβ (Fig. 4B). Careful analysis of the commonly deregulated gene expressions in each comparison revealed a specific "Granulosa signature". More precisely, GSEA leading edge comparisons showed that 116 genes were upregulated, while 132 genes were downregulated in LXR DKO compared to the two other genotypes (Fig. 4C, Dataset EV4). Hierarchical clustering revealed two clusters of genes with a highly significant signature closely related to mouse LXR status (Fig. 4D), confirming that these genes are regulated by PMSG in granulosa cells but display an altered expression in LXR DKO ovaries (Fig. 4E). Interestingly, a gene that displays repressed expression by PMSG were found upregulated in LXR DKO ovaries. Conversely, gene expression stimulated by PMSG showed a repression profile in LXR DKO ovaries (Fig. 4E).

Among the most deregulated genes, Txnip gene encoding the Thioredoxin Binding Protein, initially identified as an actor of thioredoxin activity and cell oxidative potential, was particularly interesting given its expression increase in LXR DKO ovaries (Figs. 4D,E and 5A). Indeed, TXNIP accumulation is strongly enriched in granulosa cell (Fig. 5B). TXNIP accumulation is tightly correlated with inflammation occurring 40 h post-stimulation since accumulation remains low at 24 h post-stimulation (Appendix Fig. S7). In line with p65 detection in LXR DKO ovaries, TXNIP accumulation is restricted to mural granulosa (Fig. 5B) suggesting that with compartment is critical to orchestrate inflammatory response. As PSMG inducing downstream AMPc/PKA signaling pathway in granulosa cells, we confirmed that TXNIP is directly regulated by this pathway using the pharmacologic PKA activator forskolin (Fig. 5C,D). In human KGN and mouse primary granulosa culture, this stimulation repressed TXNIP expression. In parallel, similar experiments using DKO mouse granulosa primary culture showed that, even though forskolin treatment still repressed Txnip expression, the LXR-null context led to a significantly relapsed treatment repression (Fig. 5D). In contrast, experiments performed with TgAMH-LXRβ rescued mouse cells recapitulated the wild-type model (Fig. 5D). Together, these findings highlight that upregulation of Txnip observed in vivo in LXR DKO mice is granulosa-cell autonomous. We then questioned whether gene deregulation resulted in protein level deregulation. Indeed, the analysis of the TXNIP-immunolabelled sections showed staining extended to the whole thickness of the granulosa in the DKO mice compared to the wild-type and TgAMH-LXRβ mice, which presented a centrifugal gradient labeling (Fig. 5B). Finally, we assessed the same possible link

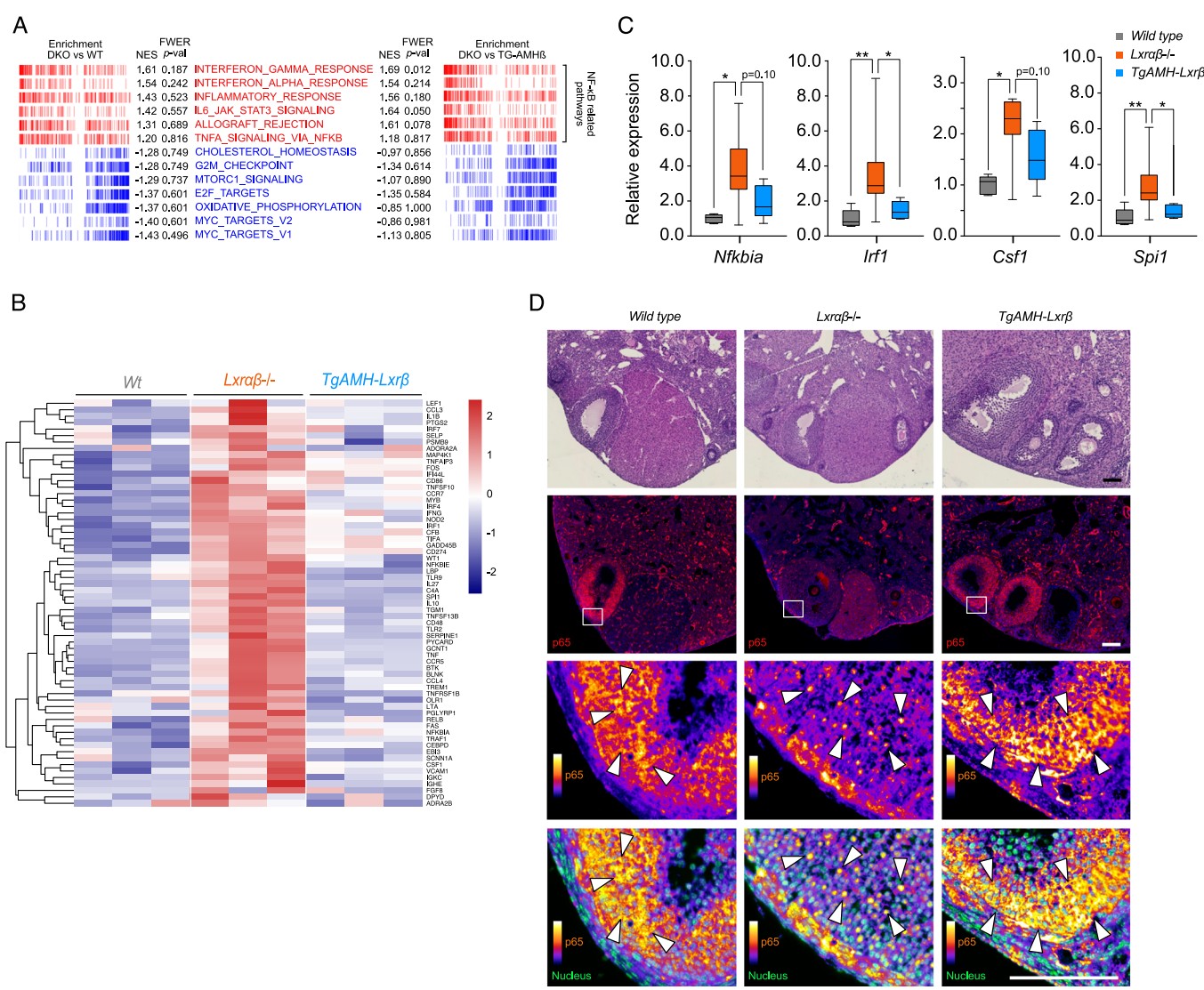

**Figure 3. LXR DKO ovaries showed an alteration of NF-κB signaling response.**

(A) GSEA comparisons of wild type versus LXR DKO and LXR DKO versus TG-AMH-Lxrβ. Categories in red indicate NF-κB pathway deregulations. (B) Heatmap of NF-κB targets genes (Boston University list) using 40 h post-PMSG dataset. (C) *Nfkbia, Irf1, Csf1*, and *Spi1* gene expression analyzed by RT-qPCR on wild type $n = 7$, LXR DKO $n = 7$, and TG-AMH-Lxrβ $n = 5$ ovaries. (D) HE-staining (upper panel) and p65 immunodetection in red (middle panel), magnification of granulosa compartment with p65 staining using fire scale and nucleus in green (bottom panel). Wild type and TG-AMH-Lxrβ ovaries showed a cytoplasmic staining compared to LXR DKO that exhibit a nucleus translocation of p65 in yellow (white arrows) (Scale bars = 100 μm). In (C), boxes extend from the 25th to 75th percentile, the middle line shows the median, whiskers extend to the most extreme data. In (C), averages values ± SD are represented. Significance determined in (C) by Mann and Whitney test. *$P < 0.05$, **$P < 0.01$ (exact *P* values for these statistical comparisons are shown in Appendix Table S1). Source data are available online for this figure.

in humans. RNA single cell analysis shows that expressions of *TXNIP* and *NR1H2*, encoding LXRβ, are tightly correlated during folliculogenesis in human ovary (Fig. 5E) (Zhang et al, 2018). Thus, we uncovered a granulosa cell-specific gene signature in response to PMSG, that is altered by the lack of LXR and identified TXNIP as a possible candidate to support OHSS phenotype.

## TXNIP/inflammasome axis in granulosa cells controls ovary inflammation in an LXR-dependent manner

The inflammasome is a supramolecular complex involved in the cleavage of pro-inflammatory precursor cytokines, pro-IL1β and

pro-IL18, into mature forms, IL1β and IL18, respectively. Regulation of the inflammasome is a two-step process (Swanson et al, 2019). The first step, called priming, is characterized by an increase in the gene expression of inflammasome components such as *Nlrp3, Asc, Casp1*, and *Il1b*, mainly orchestrated by the NFκB signaling. The second step consists of inflammasome oligomerization, called activation, and is under the control of a wide range of intracellular stimuli. In 2010, a new role of TXNIP emerged as an important regulator of inflammasome activation (Zhou et al, 2010). Neutrophils influx as well as IL1β release in intraperitoneally stimulated animals with monosodium urate crystals, potent inducers of NLRP3-inflammasome, is strongly impaired in *Txnip-/-* mice,

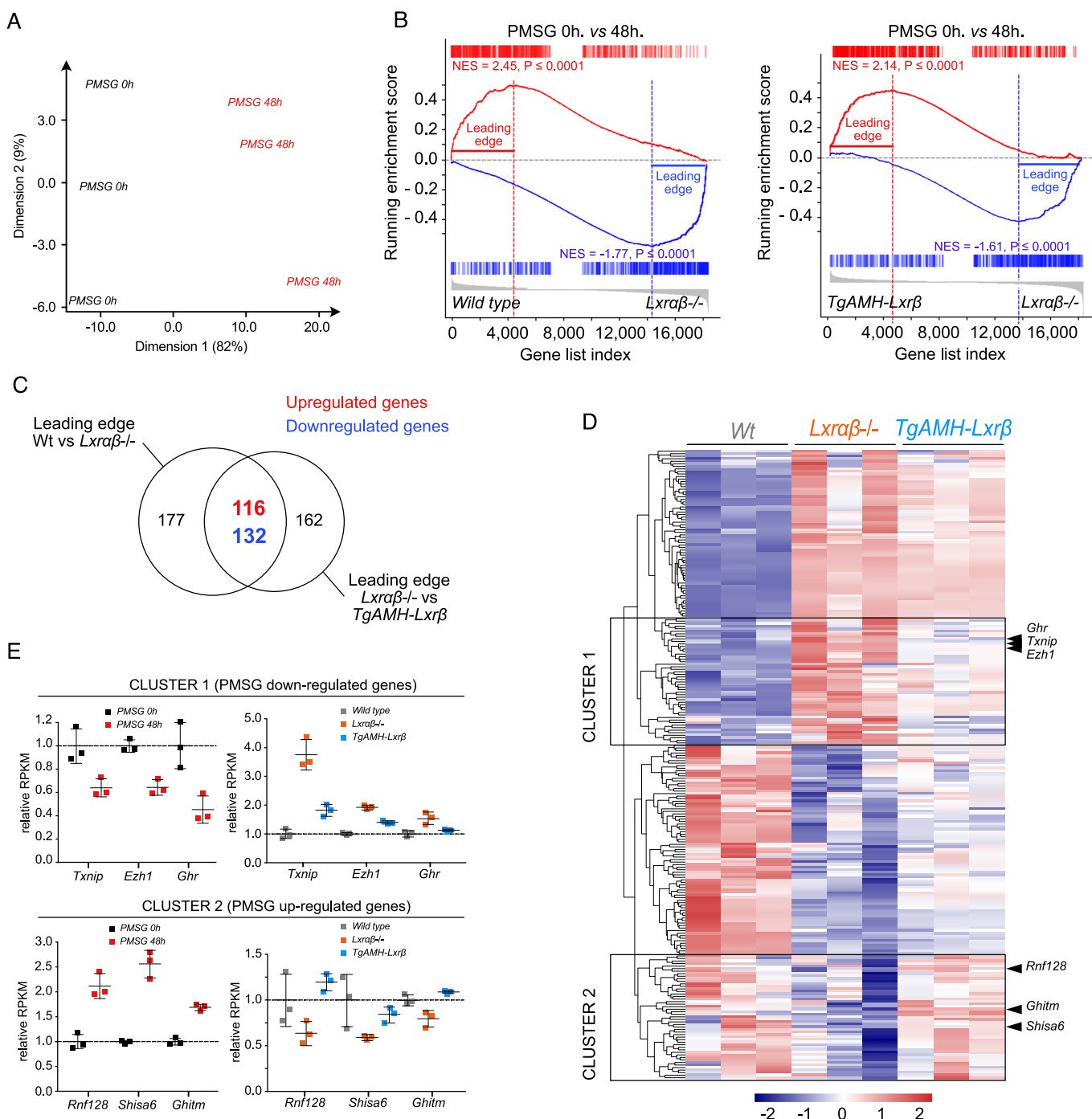

**Figure 4. LXR DKO mice reveals an impaired response to PMSG stimulation specific to Granulosa cells.**

(A) Principal component analysis from granulosa cell collected on immature mice before and after PMSG stimulation for 48 h (Madogwe et al, 2020). (B) GSEA from PMSG-responders granulosa specific gene list using wild type versus LXR DKO (left panel) and TG-AMH-Lxrβ versus LXR DKO (right panel) comparison. (C) Venn diagram identifying leading-edge genes deregulated in both comparisons. (D) Heatmap of the 248 expression gene profiles identified as PMSG-responders of granulosa cells from wild type, LXR DKO, and TG-AMH-Lxrβ mouse ovaries. (E) Gene expression profiles of *Txnip*, *Ezh1*, and *Ghr* (Cluster 1) and *Rnf128*, *Shisha6*, and *Ghitm* (Cluster 2) in granulosa cells PMSG-stimulated 0 h n = 3 vs. PMSG-stimulated 48 h n = 3 compared to 40 h post-PMSG induced wild type n = 3, LXR DKO n = 3 and TG-AMH-Lxrβ n = 3 ovaries dataset.

supporting a key role of TXNIP in activation step. Conversely, TXNIP overexpression through lentiviral transduction in THP1 cells leads to significantly sensitization of the inflammasome to various known inducers. Considering this connection between

TXNIP and inflammasome, we first explored expression profiles of genes encoding inflammasome components, such as *Nlrp3*, *Pycard*, *Il1b*, and *Casp1*, together with *Txnip* (Fig. 6A). We observed that all of them were upregulated in LXR DKO mice compared to both

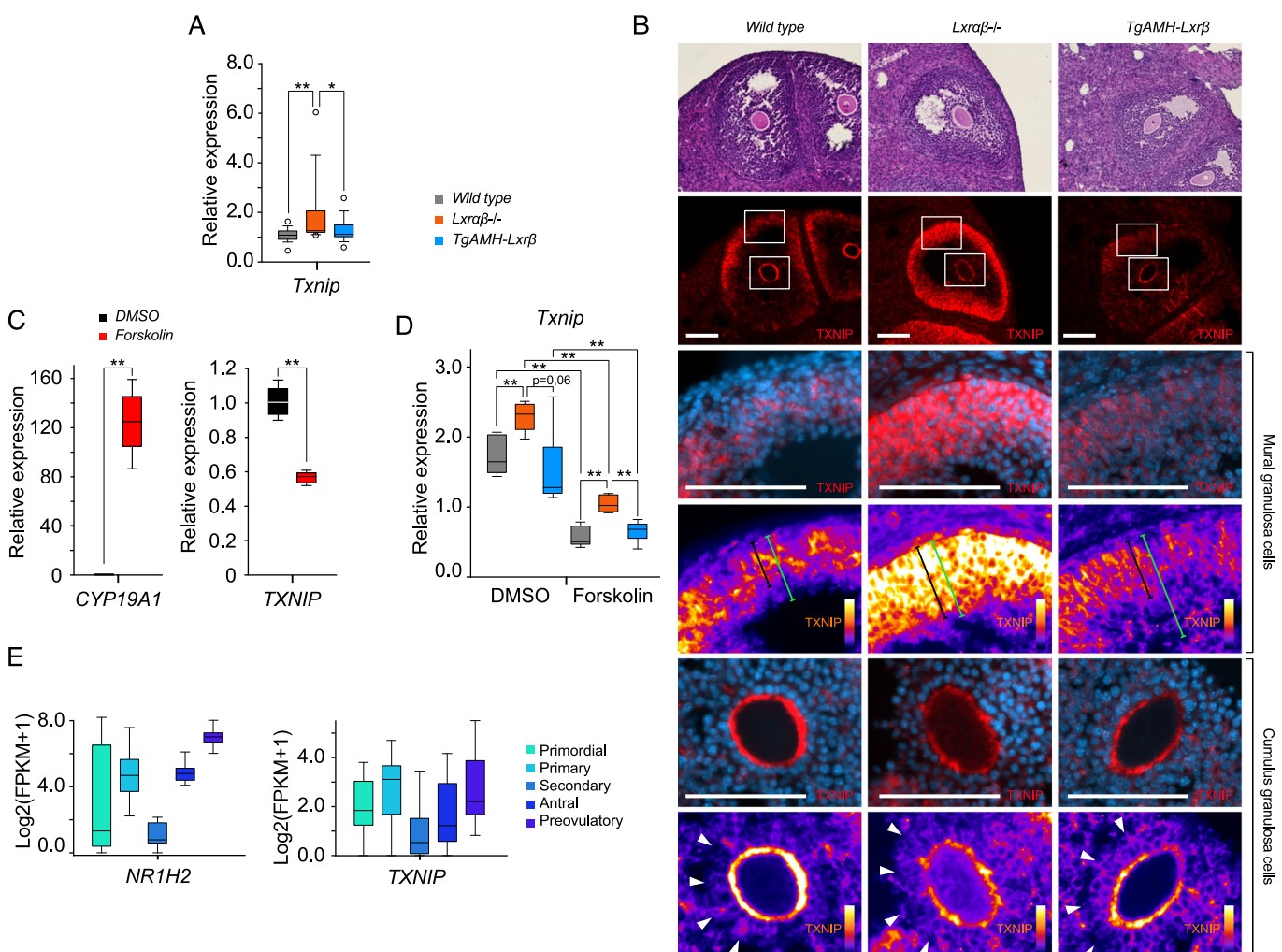

**Figure 5. *Txnip* expression is under the control of both PMSG and LXR activity.**

(A) *Txnip* expression analysis by RT-qPCR in wild type $n = 18$, LXR DKO $n = 14$, and TG-AMH-Lxrβ $n = 15$ mouse ovaries. (B) Immunodetection of TXNIP in wild type, LXR DKO, and TG-AMH-Lxrβ ovaries in red (upper panel, scale bar = 1mm; middle panel, scale bar = 100 μm), high magnification reveals that TXNIP is accumulated is the entire mural granulosa layer in LXR DKO compared to both wild type and TG-AMH-Lxrβ that harbor a centrifuge expression gradient pattern (black brackets: TXNIP intensity signal vs green brackets: granulosa layer). TXNIP detection did not harbor any difference in cumulus granulosa cells whatever the genotype (scale bar = 100 μm). (C) Gene expression analysis of *TXNIP* and *CYP19A1* in KGN human granulosa cell line in response to DMSO $n = 6$ or forskolin $n = 6$ treatment. (D) *Txnip* expression in primary granulosa cell cultures from wild type, LXR DKO, and TG-AMH-Lxrβ mouse ovaries in response to forskolin treatment ($n$ for each comparison are shown in Appendix Table S1). (E) Single-cell RNAseq box plots of both *TXNIP* and *NR1H2* expression from granulosa cell originated from follicle at various stage of maturation. In (A), (C), and (D), boxes extend from the 25th to 75th percentile, the middle line shows the median, whiskers extend to the most extreme data. In (A), (C) and (D), averages values ± SD are represented. Significance determined in (A), (C), and (D), by Mann and Whitney test. \*$P < 0.05$, \*\*$P < 0.01$ (exact $P$ values for these statistical comparisons are shown in Appendix Table S1). Source data are available online for this figure.

wild-type and TgAMH-LXRβ mice (Fig. 6B), resulting in increased protein accumulations (Fig. 6C), indicating that LXR ablation leads to enhanced inflammasome priming. Moreover, at the transcriptional level, *Nlrp3*, *Asc,* and *Il1b* are closely correlated (Fig. 6D), strongly suggesting a coordinated gene regulation network. To assess involvement of inflammasome in the ovarian hemorrhagic phenotype exhibited by LXR DKO mice in response to hormonal stimulation, we treated these mice with MCC950, a selective NLRP3 inhibitor. MCC950 treatment started 24 h prior the first PMSG injection and continued every day throughout the stimulation protocol (Fig. 6E). As a result, we observed a significant decrease in the number of hemorrhagic cysts in LXR DKO mice. This latest

finding definitively shows that OHSS-like phenotype in LXR DKO mice is dependent of inflammasome. To localize and track down inflammasome activity, we detected IL1β accumulation in the ovary LXR DKO mice (Fig. 6F). Indeed, IL1β accumulation is observed in the stroma due to immune cells secretion (ROI1, Fig. 6F). Granulosa cells showed two distinct stainings. It is nearly negative in follicle harboring normal size (ROI2, Fig. 6F), or positive cells are found in mural granulosa with an important accumulation in the antrum of enlarged follicles, indicating IL1β accumulation in follicle fluid (ROI3, Fig. 6F). These observations show that follicles that present a pathological response to PMSG specifically display an inflammatory profile with a high rate of pro-inflammatory

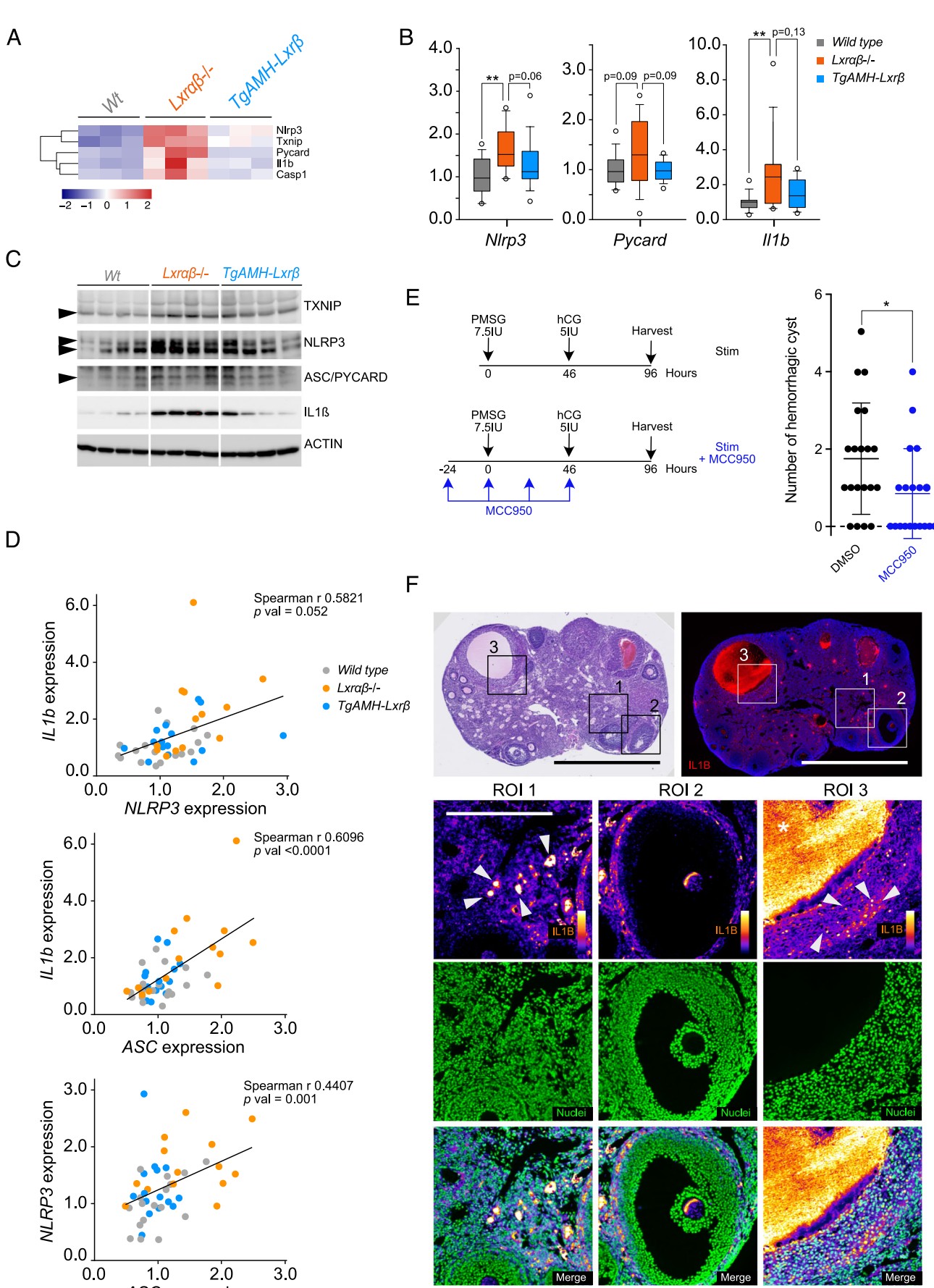

◀  **Figure 6.  Impaired inflammasome activity triggers hemorrhagic phenotype observed in LXR DKO.**

(A) Heatmap of *Nlrp3, Txnip, Pycard (Asc),* *Il1b,* and *Casp1* using 40 h post-PMSG dataset. (B) *Nlrp3, Pycard (Asc),* and *Il1b* expression analysis by RT-qPCR in wild type $n = 17$, LXR DKO $n = 16$, and TG-AMH-Lxrβ $n = 15$ mouse ovaries. (C) TXNIP, NLRP3, ASC, IL1B protein accumulation in wild type, LXR DKO, and TG-AMH-Lxrβ mouse ovaries. ACTIN was used as a loading control. (D) Correlation plots between NLRP3, ASC, and IL1B using RT-qPCR data from in wild type, LXR DKO, and TG-AMH-Lxrβ mouse ovaries. (E) Protocol for hormonal stimulation as described Fig. 1A with $n = 19$ or without MCC950 $n = 20$ co-treatment, an inflammasome inhibitor. MCC950 was injected every 24 h, by four injections starting 24 h prior first PMSG injection (7.5 UI) until hCG final injection (5IU). Hemorrhagic cysts have been quantify comparing DMSO, as a control, and MCC950 treatment. (F) HE-staining (left panel) and IL1B immunodetection (right panel) were performed using LXR DKO ovary (Scale bar = 1 mm). Magnification indicated IL1B staining, using fire scale, and nucleus in green. Strong staining has been observed (white arrows, white asterisk) both surrounding immune cells (ROI 1) as well as cystic follicle in the antrum (ROI 3) compared to mural granulosa cells of non-cystic follicle (ROI 2) (Scale bar = 100 μm. In (B), boxes extend from the 25th to 75th percentile, the middle line shows the median, whiskers extend to the most extreme data. In (E), averages values ± SD are represented. Significance determined in (B) and (E), by Mann and Whitney test. *$P < 0.05$, **$P < 0.01$ (exact $P$ values for these statistical comparisons are shown in Appendix Table S1). Source data are available online for this figure.

cytokines production. Altogether, these results provide evidence that the control of the inflammasome by granulosa cells' LXR is a pivotal crossroad to control inflammation processes in the follicle during hormonal stimulation.

## Immune signature persists in cumulus granulosa cells post-ovulation and unmask immune cell infiltration retrieves in patients with PCOS

Assisted reproductive technology is a routinely clinical procedure implemented worldwide. Nevertheless, research in this field is strictly regulated, particularly due to the ethical concerns surrounding the manipulation of the human embryo. Here, we have taken advantage that, during ART procedures, the granulosa cells of the cumulus that surround oocytes are dissociated and eliminated. Thus, these cells are considered as surgical waste and can be harvested for scientific research purposes following patient's consent. Using cells from control and PCOS patients (Fig. 7A; Appendix Fig. S8, Dataset EV5), we explore cumulus granulosa RNAseq datasets. This analysis was carried out in parallel using cumulus cells originated from wild-type, LXR DKO, and TgAMH-LXRβ transgenic models (Fig. 7A,B). The comparison between human and mouse shows that an enrichment of the immune/inflammatory signature present both datasets (Fig. 7B). In line with the restricted activation of NF-κB pathway and TXNIP accumulation in mural granulosa (Figs. 3D and 5B), inflammasome signature is not conserved in cumulus granulosa cells either in mice than in PCOS patients (Appendix Fig. S9). Next, we compared gene leading edge of each pathway that was identified both using PCOS cohort and the comparison wild-type *vs* LXR DKO and TgAMH-LXRβ *vs* LXR DKO in cumulus datasets (Dataset EV6). Carefully analysis and crossing differentially expressed genes (DEGs) leads to identify the hyperstimulation gene signature (HPS signature) that aggregate 96 genes (Fig. 7C; Appendix Fig. S10). Focus on this HPS signature reveals a significant deregulation in PCOS patients (Fig. 7D,E) particularly enriched with immune associated genes (Fig. 7F). Analysis of the gene expressions that composed HPS signature using single-cell RNAseq dataset from human ovary (Zhang et al, 2018) shows that significant numbers are specific to the immune cell compartment (Fig. 7G) whose deregulation were confirmed by RT-qPCR (Appendix Fig. S11). Among them we found *PTRC* encoding CD45, a pan-leukocyte marker, as well as *TNFRSF1B*, TNF-receptor superfamily member, already identify as a DEGs in LXR DKO mouse ovary (Fig. 3B). Together, these findings revealed putative infiltration of COC by immune cells. Indeed, we observed

that post-ovulation, LXR DKO showed a strong CD45+ staining in hemorrhagic cysts thus suggesting the accumulation of immune cells in the antrum in mature follicles (Appendix Fig. S12). Such observation prompted us to analyze such phenomenon inside the cumulus oocyte complex in LXR DKO transgenic mouse cumulus. We performed CD45 staining cumulus-oocyte complex after ovulation on wild-type, LXR DKO, and TgAMH-LXRβ mice. We identified an increase in immune cell infiltration in LXR DKO cumulus-oocyte complex compared to WT and TgAMH-LXRβ mice (Fig. 7H; Appendix Fig. S13A,B). To confirm the significance of the HPS signature, we analyzed single-cell RNAseq statset from a PCOS model induced by chronic exposure to DHEA (Appendix Fig. S14A–D) (Luo et al, 2024). We focused our attention on the immune cell population in (Appendix Fig. S15A,B). Analysis of the immune cell cluster revealed that the HPS signature is present in most of cells (Fig. 7I). Unexpectedly, the HPS signature is downregulated in the immune cells of the PCOS model compared to patients. This discrepancy could be explained by hormonal differences, as PCOS patients are typically stimulated for ovulation, whereas PCOS mice remain in a basal anovulatory state. Thus, immune cells in the PCOS model exhibit similarities to those in the LXR DKO model in terms of deregulated gene panels. Beyond characterizing an abnormal response to ovarian stimulation, the HPS signature also reveals an intrinsic predisposition gene expression program in the PCOS context, highlighting its increasing clinical relevance. Altogether, these results indicate that the HPS signature identified by the comparison between preclinical mouse models and PCOS patient dataset is tightly associated with the pathological response to hormonal stimulation and PCOS status. These finding open the field to identify relevant biomarkers to predict abnormal response in PCOS patients.

## Discussion

Here, we report that pharmacological targeting of LXR during hormonal stimulation could reduce phenotypic traits related to hemorrhagic outcomes and molecular markers. The molecular etiology involves the central coordination of gonadotrophin response orchestrated by the granulosa cells during folliculogenesis. LXR are require for the regulation of the inflammasome activity in the ovary, which is necessary to fine tune the inflammation steady state level during the ovulation. As a result, LXR dysfunction leads to an imbalance in the inflammation process that triggers a massive immune infiltration that remains present in oocytes-cumulus complexes after ovulation.

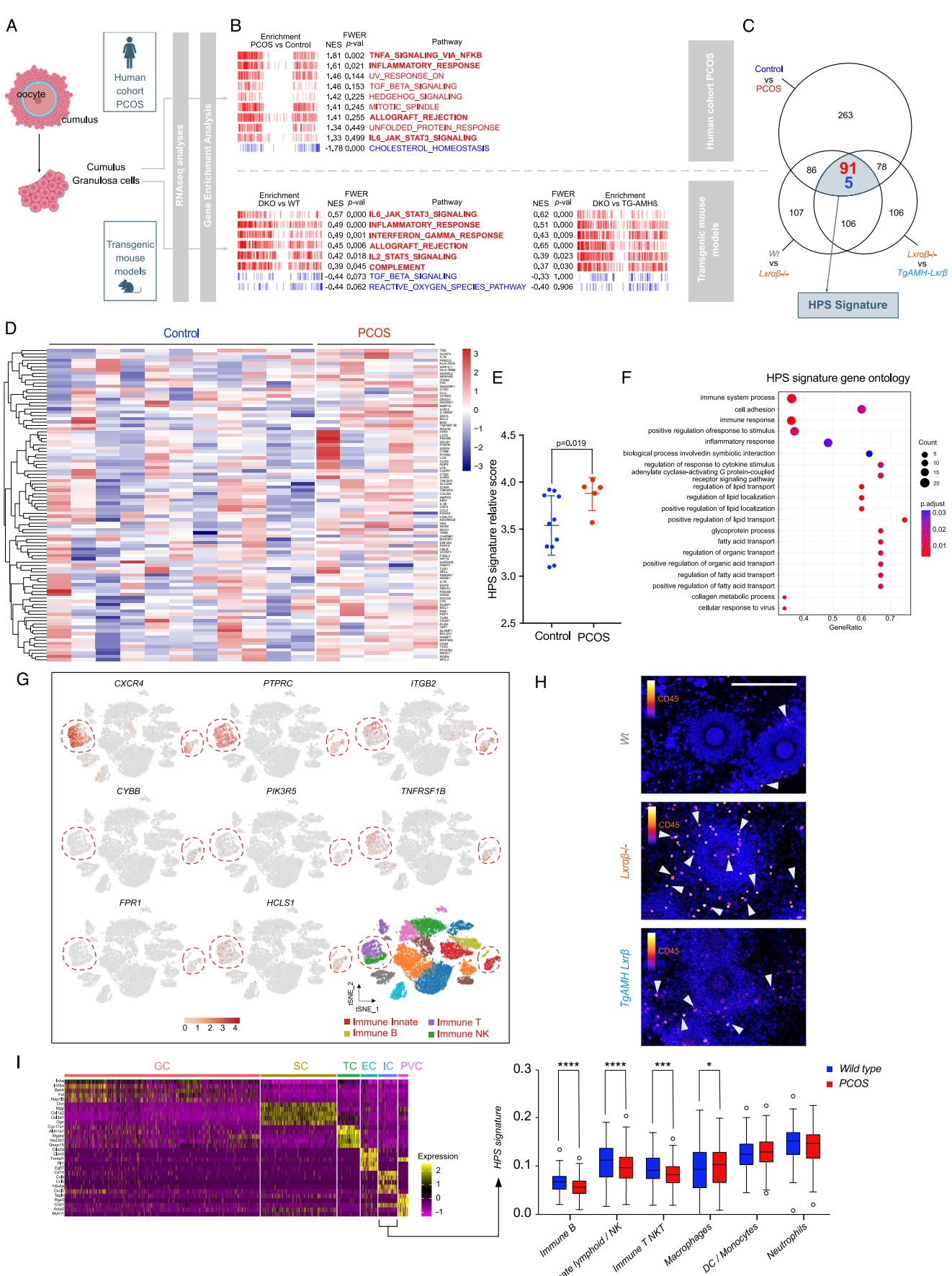

**Figure 7. Inflammatory signature in cumulus oophorus reveals an immune infiltration both in human and mouse models in a context of PCOS.**

(A) Granulosa cells were collected after COC hyaluronidase treatment and process for subsequent RNAseq library and sequencing. (B) GSEA comparisons of human cohort PCOS versus Control as well as mouse models, LXR DKO versus wild type and LXR DKO versus TG-AMH-Lxrβ. Categories are red indicated upregulated and is blue downregulated pathways. Bold pathway is associated with immune signature. (C) Venn diagram comparing common leading-edge lists of genes between human and mouse datasets, this analysis leads to identify 96 DEGs that composed the HPS signature. (D) Heatmap corresponding to the HPS signature. (E) HPS signature significance calculation using GSVA between human cohort PCOS n = 5 versus Control n = 11 samples. (F) Gene ontology analysis using Cluster Profiler obtained from human cohort PCOS versus Control regarding the HPS signature. (G) Human adult ovary single-cell RNAseq analysis. CXCR4, PTRC, ITGB2, CYBB, PIK3R5, TNFRSF1B, FPR1, IL18RAP, and HCLS1 are plotted using tSNE clustering. Immune cell clusters have been indicated using red dashed lines. (H) CD45 immunodetection of immune cells in the cumulus-oocyte complex from wild type, LXR DKO, and TG-AMH-Lxrβ (white arrows, scale bar = 100 μm). (I) HPS signature analysis on single cell cluster of immune cells in a murine model of PCOS (Luo et al, 2024). In (E) and (I), average values ± SD are represented. Significance determined in (E), by Mann and Whitney test. *P < 0.05, ***P < 0.001, ****P < 0.0001 (exact P values for these statistical comparisons are shown in Appendix Table S1). Source data are available online for this figure.

Our study shows that the ovary stimulation is involved in the control of inflammasome activity in mural granulosa cells and is supported by LXR. Indeed, the use of the inflammasome inhibitor MCC950 makes it possible to compensate for the absence of LXR and restore a proportional hormonal response in DKO LXR mice. The natural homeostasis of the ovary requires the mobilization of inflammatory processes, particularly at the time of ovulation and the inflammasome plays an important role throughout the ovarian cycle. However, chronic low inflammation noise can be detrimental to the proper functioning of the ovary. Asc-/- and Nlrp3-/- mice exhibit reduced inflammation with decreased production of pro-inflammatory cytokines Il6, Il1β, and TNFα, leading to a delay in the depletion of the follicular reserve and a lengthening of the window of fertility (Lliberos et al, 2020). In addition, abnormal activation of the inflammasome is observed in patients with PCOS (Lai et al, 2022; Liu et al, 2021). Our study has some limitations since it is restricted to cumulus granulosa cells. It would be interesting to study the activation status of LXR and that of inflammasome in mural granulosa cells collected in the follicular fluid during oocyte puncture from PCOS and control patients. Thus, the regulatory mechanisms of the inflammasome occupy a crucial place in ovarian homeostasis. Pharmacological targeting of LXR to control inflammasome may be an attractive option from a clinical point of view, even though synthetic agonists of LXRs have some significant side effects.

PMSG is a bipotential hormone capable of binding and activating both follicle stimulating hormone receptor (FSHR) and luteinizing hormone/choriogonadotropin receptor (LHCGR) (Licht et al, 1979). The induction window that we targeted 40 h post-injection of PMSG corresponds to a phase of pre-luteinization of granulosa cells within the follicle, characterized by a strong increase in the expression of LHCGR. It is therefore possible that the OHSS-phenotype observed in LXR DKO mice is dependent on both follicle stimulating hormone (FSH) and luteinizing hormone (LH) signaling. This point requires further investigation to decide on the respective involvement of each of these pathways.

According to our results, TXNIP stands as a key crossroads in the integration of hormonal signals and the control of the inflammatory status of the ovary. Indeed, TXNIP is both a target of inflammasome priming processes, being part of genes putatively upregulated by stimulation of the NF-κB pathway, and in the activation mechanism by controlling the activity of the inflammasome itself (Zhou et al, 2010). The molecular link of TXNIP with the inflammasome and its deregulation in pathological conditions show the key role that this actor plays in the control of inflammation within the ovary. The control mechanism of Txnip expression by LXR that leads to the partial loss of PMSG-induced

inhibition remains to be elucidated. No potential LXR binding site (DR-1 or DR-4) has been identified into the promoter so far. However, Txnip is a strong candidate to be controlled by the NF-κB pathway. Given the repressive control exerted by LXR on the NF-κB signaling, it is possible that their invalidation supports of the overexpression of Txnip.

The phenotype of LXR DKO mice following hormonal stimulation for superovulation indicates the key role of LXR signaling in granulosa cells during folliculogenesis. The question therefore is to identify the endogenous ligands able to control the activity of these receptors. Thus, the Follicular fluid meiosis-activating sterol (FF-MAS) appears as an obvious candidate. Indeed, FF-MAS is an LXR ligand and is involved in GVBD for the resumption of meiosis and the expansion of the cumulus oophorus in ovary. Futhermore, the expression of the gene encoding the enzyme responsible for its synthesis, CYP51, is positively controlled by FSH and LH. This could lead to LXR-dependent feedback of the inflammation as suggested by our data. However, it remains possible that other endogenous ligands may be involved. Indeed, FSH stimulates the de novo synthesis of cholesterol (Barañao and Hammond, 1986; Guo et al, 2019), thus opening the possibility of the production of other oxidized secondary metabolites potentially able to module the activity of LXR.

As we observed in our results, the most abundant cytokine accumulated in LXR DKO mice after stimulation is IL1β. Several studies report the overexpression of this interleukin in cases of PCOS (Liu et al, 2021; Qi et al, 2019) with an increased and measurable accumulation in the serum of patients. Currently, the blood estradiol (E2) assay is the reference marker for monitoring the response to hormonal stimulation in patients. It would therefore be of interest, in parallel with E2, to monitor serum levels of Il1β to obtain additional information on the inflammatory state of the ovary. Similar observations have been done for TXNIP (Wu et al, 2014). Together, these assays using IL1β and/or TXNIP as a marker could thus provide a powerful tool to detect early abnormal response to hormonal stimulation during ART procedures.

In conclusion, this study shows that LXR in a central integrator for gonadotrophin responses in the ovary during artificial hormonal stimulation. We showed that synthetic LXR ligand such as GW3965 can prevent ovarian hyperstimulation. We uncovered that LXR exert this role mainly through their actions within the granulosa cells by the control of inflammation. This control is the turning point before pathological response mainly sustained by an abnormal immune infiltration. Targeting LXR as a modulator of hormonal stimulation during ART protocols, especially in PCOS patients, could be a relevant therapeutic pathway for women with high risks of hyperstimulation.

# Methods

**Reagents and tools table**

| Reagent/Resource | Reference or Source | Identifier or Catalog Number | | | | |
|---|---|---|---|---|---|---|
| **Experimental models** | | | | | | |
| Wild type | Dr. David Mangeldorf's Lab (Department of Pharmacology and Biochemistry, University of Texas Southwestern Medical Center, Dallas, TX) | | | | | |
| LXRαβ-/- | Dr. David Mangeldorf's Lab (Department of Pharmacology and Biochemistry, University of Texas Southwestern Medical Center, Dallas, TX) | | | | | |
| TgAMH-Lxrβ | See reference Maqdasy et al, 2015 in the main text | | | | | |
| NOD-SCIDγ (NSG) | Charles River | 614NSG | | | | |
| KGN cell line | See reference Nishi et al, 2001 in the main text | | | | | |
| **Antibodies for immunohistochemistry** | | Clone | Reference | Antigen retrival | Blocking | Dil. |
| CD45 | BD Pharmingen | 30-F11 | 550539 | Without | Horse serum 2.5% | 1/200 |
| ENDOMUCIN | Clinisciences | V.7C7 | sc-65495 | Citrate Tween 0.05% | BSA 1% | 1/500 |
| MHCII | Invitrogen | M5/114.15.2 | 467561 | without | Horse serum 2.5% | 1/200 |
| P65 | Cell Signaling | D14E12 | 8242 | Citrate pH6 | TBS-tween 0.1%-NGS 5% | 1/1000 |
| TXNIP | Abcam | EPR14774 | 188865 | Tris EDTA | Horse serum 2.5% | 1/10,000 |
| **Antibodies for western blot** | | Clone | Reference | MW (kDa) | Dil. | Host |
| ACTIN | Sigma-Aldrich | | A2066 | 42 | 1/5000 | Rabbit |
| ASC/PYCARD | Novus Biologicals | | NBP1-78977 | 22 | 1/1000 | Rabbit |
| IL1B | R&D Systems | IL-IF2 | AF-401-NA | 17 | 1/800 | Mouse |
| NLRP3 | Novus Biologicals | | NBP1-77080 | 110 | 1/1000 | Rabbit |
| TXNIP | Novus Biologicals | JY2 | NBP-54578 | 55 | 1/1000 | Mouse |
| **Oligonucleotides and other sequence-based reagents** (mm: mus musculus, hs: homo sapiens) | | Primer forward | Primer reverse | | | |
| mm36b4 | Eurogentec | GTCACTGTGCCAGCTCAGAA | TCAATGGTGCCTCTGGAGAT | | | |
| mmCsf1 | Eurogentec | GCCTGTGTCCGAACTTTCCA | GGGTGGCTTTAGGGTACAGG | | | |
| mmCyp19a1 | Eurogentec | CGGAAGAATGCACAGGCTCGAG | CGATGTACTTCCCAGCACAGC | | | |
| mmFshr | Eurogentec | GTGCTCACCAAGCTTCGAGTCAT | AAGGCCTCAGGGTTGATGTACAG | | | |
| mmIl1b | Eurogentec | GCTGAAAGCTCTCCACCTCA | TGGGTGTGCCGTCTTTCATT | | | |
| mmInha | Eurogentec | TCCTGGTAGCCCACACTAGG | GAAACTGGGAGGGTGTACGA | | | |
| mmIrf1 | Eurogentec | AAGGATGCCTGTCTGTTCCG | CTTCCTCGATGTCTGGCAGG | | | |
| mmNfkbia | Eurogentec | GAGACCTGGCCTTCCTCAAC | TCTCGGAGCTCAGGATCACA | | | |
| mmNlrp3 | Eurogentec | AGAGTGGATGGGTTTGCTGG | CGTGTAGCGACTGTTGAGGT | | | |
| mmPycard | Eurogentec | GTCCCTGCTCAGAGTACAGC | TTGTCTTGGCTGGTGGTCTC | | | |
| mmSpi1 | Eurogentec | GCAGCGATGGAGAAAGCCAT | GCACCATGGGAGTATCGAGG | | | |
| mmTxnip | Eurogentec | AGTTACCCGAGTCAAAGCCG | TTCCAGGCCTCATGATCACC | | | |
| hsCXCR4 | Eurogentec | GGGCAATGGATTGGTCATCCT | TGCAGCCTGTACTTGTCCG | | | |
| hsCYBB | Eurogentec | ACCGGGTTTATGATATTCCACCT | GATTTCGACAGACTGGCAAGA | | | |

| Reagent/Resource | Reference or Source | Identifier or Catalog Number | |
|---|---|---|---|
| hsCYP19A1 | Eurogentec | AAATGCTGATCGCAGCTCCT | CTGGTACCGCATGCTCTCAT |
| hsFPR1 | Eurogentec | TGGGAGGACATTGGCCTTTC | GGATGCAGGACGCAAACAC |
| hsHCLS1 | Eurogentec | AGTGGGCCATGATGTGTCTG | CTCCCCATCGTTGCTCCTTT |
| hsIL18RAP | Eurogentec | ATGCTCTGTTTGGGCTGGATA | GTGAGAGTCGATTTCTGTGGC |
| hsITGB2 | Eurogentec | TTCGGGTCCTTCGTGGACA | ACTGGTTGGAGTTGTTGGTCA |
| hsPIK3R5 | Eurogentec | GGGAGGCTGTTCCTCTAACAC | GTTCACGGAGGTACAGACCTT |
| hsPTPRC | Eurogentec | ACCACAAGTTTACTAACGCAAGT | TTTGAGGGGGATTCCAGGTAAT |
| hsTNFRSF1B | Eurogentec | TGAAACATCAGACGTGGTGTG | TGCAAATATCCGTGGATGAAGTC |
| hsTXNIP | Eurogentec | TGTTCCCGAATTGTGGTCCC | GGATGTTGCAGCCCAGGATA |
| **Chemicals, Enzymes and other reagents** | | | |
| Pregnant Mare Serum Gonadotroin | Ceva | SYNCRO-PART PMSG 600 | |
| Human chorionic Gonadotropin | MSD | CHORULON 1500 | |
| GW3965 | Sigma-Aldrich | G6295 | |
| Methyl cellulose | Sigma-Aldrich | M7027 | |
| MCC950 | Selleckchem | S7809 | |
| DMSO | Sigma-Aldrich | D8418 | |
| ImmPress Polymer Detection Kit | Vector Laboratories | 101098-258 101098-260 101098-262 | |
| SuperBoostTM Kits with Alexa Fluor 555 Tyramide for fluorescence | Invitrogen | B40923 | |
| Hoechst | Invitrogen | H3570 | |
| Phenylmethylsulfonyl fluoride | Sigma-Aldrich | 10837091001 | |
| Complete protease inhibitors | Roche Molecular Biochemicals | 4693116001 | |
| NaF | Sigma-Aldrich | S1504 | |
| $Na_3VO_4$ | Sigma-Aldrich | 567540 | |
| Trans-Blot® Turbo™ nitrocellulose membranes | Bio-Rad | 1704271 | |
| Peroxidase-conjugated anti-rabbit antibody | Abliance | BI 2407 | |
| Peroxidase-conjugated anti-mouse antibody | Abliance | BI 2413C | |
| Clarity™ Western ECL Blotting Substrates | Bio-Rad | 1705060 | |
| Clarity Max™ Western ECL Blotting Substrates | Bio-Rad | 1705062 | |
| Evans blue | Sigma-Aldrich | E2129 | |
| Formamide | Sigma-Aldrich | F7503 | |
| Trizol® Reagent | Invitrogen | 15596018 | |
| Nucleospin RNA L extraction kit | Macherey Nagel | 740955.250 | |
| Random primers | Promega | C1181 | |
| Reverse transcriptase (M-MLV RT) | Promega | M1701 | |
| RNAsin | Promega | N2615 | |
| SYBR qPCR Premix Ex Taq II Tli RNase H+ | Takara | TAKRR820W | |
| Superscript IV | ThermoFisher Scientific | 18090010 | |

| Reagent/Resource | Reference or Source | Identifier or Catalog Number |
|---|---|---|
| DMEM/HamF12 medium | Invitrogen | 31331-028 |
| L-glutamine | Invitrogen | 25030081 |
| Penicillin/Streptomycin | Invitrogen | 15140148 |
| FBS | Eurobio | CVFSVF00-01 |
| HEPES | Invitrogen | 15630080 |
| EGTA | Sigma-Aldrich | E9884 |
| BSA | Sigma-Aldrich | 05470 |
| M-199-H | Sigma-Aldrich | M7528 |
| Goat Anti Rat Alexa 568 | Invitrogen | ab175476 |
| Red blood lysis buffer | Abcam | ab204733 |
| Anti-CD16/CD32 | Pharmigen | 553141 |
| Live/Dead fixable green dead cell stain kit | Invitrogen | L34969 |
| Anti-CD19 APC clone 6D5 | Biolegend | 115512 |
| Isotype control clone K2758 | Biolegend | 400512 |
| **Software** | | |
| IMARIS | Oxford Instruments | v10.0.1 |
| cutadapt | See reference Martin, 2010 in the main text | v3.2 |
| FASTQC | Babraham Bioinformatics | v0.11.7 https://www.bioinformatics.babraham.ac.uk/projects/fastqc/ |
| Hisat2 | See reference Kim et al, 2019 in the main text | v2.2.1 |
| R/R studio | Posit Software | v2024.04.1 + 748 |
| Deseq2 | See reference Love, 2014 in the main text | |
| DeepVenn | See reference Hulsen, 2022 in the main text | |
| pheatmap | Github | https://github.com/raivokolde/pheatmap |
| cluster profiler | See reference Wu et al, 2021 and Anon in the main text | |
| CIBERSORTx | See reference Newman et al, 2019 in the main text | https://cibersortx.stanford.edu |
| MSigDB | Broad Institute Inc. MIT/UC San Diego | https://www.gsea-msigdb.org/gsea/msigdb/human/collections.jsp |
| GSEA | Broad Institute Inc. MIT/UC San Diego | v4.1.0 |
| ggplot2 | See reference Wickham, 2016 in the main text | v3.5.1 |
| Ovogrowth | See reference Fan et al, 2019 in the main text | http://ovogrowth.net/ |
| Seurat | Satijalab | v4.4.0 |
| Harmony | See reference Korsunsky et al, 2019 in the main text | v1.2.3 |
| Seurat Extend | See reference Hua et al, 2024 in the main text | v1.1.4 |
| **Other** | | |
| Zeiss AxioImager with Apotome2 | Zeiss | |
| Zeiss Axioscan Z1 slide scanner | Zeiss | |
| Chemidoc MP Imaging system | Bio-Rad | |
| Zeiss 800 Airyscan for 3D confocal imaging | Zeiss | |
| Attune Nxt station | Invitrogen | |

## Animals

All experiments were approved by the Institutional Animal Care and Use Committee (#17362-201810301023962, #33602-20220309 12103457, #37414-2022051917539645). The $Lxr\alpha\beta^{-/-}$ mice were obtained from Dr. David Mangeldorf's Lab (Department of Pharmacology and Biochemistry, University of Texas Southwestern Medical Center, Dallas, TX) and were maintained on a mixed strain background (C57BL/6:129 Sv) (Peet et al, 1998). TgAMH-LXRβ mice were generated in the local transgenic facility of Génétique Reproduction et Développement as previously described (Maqdasy et al, 2015). NOD-SCIDγ (NSG) mice were purchased from Charles River Laboratories. All mice were housed in a temperature-controlled environment (22 °C ± 2 °C, 50% ± 10% humidity) with 12 h light/dark cycle (7 a.m./7 p.m.), and ad libitum access to food and water. All experiments were performed on age-matched female mice. Except when indicated, mice were 6–9 months old. For standard hormonal stimulation, mice received an intraperitoneally injection of 7 IU pregnant mare serum gonadotropin on day 1 and 5 IU human chorionic gonadotropin on day 3.

In hyperstimulation model, mice received an intraperitoneally injection of 20 IU pregnant mare serum gonadotropin on day 1 and day 2 and 10 IU human chorionic gonadotropin on day 3. 24 h prior the first PMSG injection and every day all along the hyperstimulation protocol mice were gavaged with 20 mg/kg GW3965 (Sigma-Aldrich) or vehicle (methyl cellulose). To inhibit inflammasome, 24 h prior the first PMSG injection and every day all along the stimulation protocol DKO mice were treated with intraperitoneal injection of MCC950 (Selleckchem) at 20 mg/kg daily or vehicle (DMSO) as previously described (Navarro-Pando et al, 2021).

## Histology analysis

Paraffin-embedded tissue sections were sectioned for hematoxylin and eosin staining. Immunohistochemistry was performed on paraffin-embedded tissues after antigen retrieval as indicated in Reagents and tools table, depending on the primary antibody. Tissues were blocking for 1 h with corresponding buffer as described in Reagents and tools table. Then slides were incubated overnight at room temperature with primary antibodies at the indicated concentrations. Primary antibodies were detected with appropriate polymers (ImmPress Polymer Detection Kit, Vector Laboratories). Polymer-coupled HRP activity was then detected with Tyramide SuperBoostTM Kits with Alexa Fluor 555 Tyramide for fluorescence (Invitrogen). Nuclei were counterstained with Hoechst (Invitrogen). Images were acquired with a Zeiss AxioImager with Apotome2 or Zeiss Axioscan Z1 slide scanner (Zeiss). They were minimally processed for global levels and white balance using Zeiss Zen® (Zeiss). Image settings and processing were identical across genotypes. All immunohistochemistry conditions are detailed in Reagents and tools table.

## Western blot analysis

The proteins were extracted using HEPES 20 mM, NaCl 0.42 MgCl₂ 1.5 mM, EDTA 0.2 mM, and Igepal 1% supplemented with phenylmethylsulfonyl fluoride 1 mM (Sigma-Aldrich), Complete protease inhibitors 1× (Roche Molecular Biochemicals), NaF

0.1 mM, and $Na_2VO_3$ 0.1 mM (Sigma-Aldrich). 40 µg of total protein were subjected to 10% SDS-PAGE and transferred onto Trans-Blot® Turbo™ nitrocellulose membranes. Membranes were incubated overnight at 4 °C with primary antibodies either with 5% non-fat dry milk or BSA. Primary antibody detection was performed using peroxidase-conjugated anti-rabbit or anti-mouse antibodies (Abliance) and Clarity™ or Clarity Max™ Western ECL Blotting Substrates (Bio-Rad). Antibodies used for western blots are listed in Reagents and tools table.

## Vascular permeability

Before sacrifice, mice from each group were injected under anesthesia with 100 µl of 5 mg/ml of Evans blue in the retro-orbital sinus. Ovaries were collected and incubated in 1 ml of formamide for 24 h at 60 °C. To evaluate ovarian capillary permeability, Evans blue concentration in the formamide extract was measured by light absorption at 500 and 620 nm by spectrophotometry.

## Reverse transcription qPCR

Total RNA from culture cells were extracted using Trizol® Reagent (Invitrogen) according to the manufacturer's instructions. Total RNA from frozen ovarian tissues were extracted using Nucleospin RNA L extraction kit (Macherey Nagel), according to the manufacturer's instructions. Two micrograms of total mRNAs was reverse transcribed for 1 h at 37 °C with 5 pmoles of random hexamers primers, 200 U reverse transcriptase (M-MLV RT, M1701, Promega), 2 mM dNTPs and 20 U RNAsin (N2615, Promega). Real-time quantitative PCR analysis was performed using 2 µl of 1:5 diluted cDNA template and 0.75 U of SYBR qPCR Premix Ex Taq II Tli RNase H+ (TAKRR820W, Takara). Regarding human PCOS cohort, RNA were reverse transcribed using Superscript IV (ThermoFisher Scientific) according manufacturer instructions Primer pairs are listed in Reagents and tools table. Relative gene expression was normalized to $RPLP0$ ($36b4$) using the $2^{-\Delta\Delta Ct}$ method.

## KGN cell culture and primary cultures of mouse granulosa cells

KGN cells were grown in DMEM/HamF12 medium (Invitrogen) supplemented with L-glutamine 2 mM (Invitrogen), Penicillin/Streptomycin 100 µg/mL (Invitrogen) and FBS 10% (Eurobio) at 37 °C in a humidified air 5% $CO_2$ incubator as previously described (Nishi et al, 2001). For primary granulosa cell isolation, ovaries collected from 21-day-old female mice for each genotype were sliced using a scalpel and fine forceps into sterile medium-199 plus HEPES (M-199-H, Invitrogen) containing EGTA 6.8 mM (Sigma-Aldrich), BSA 0.2% (Sigma-Aldrich), and penicillin/streptomycin 100 µg/mL. Tissue samples were incubated 10 min at 37 °C prior $250 \times g$ centrifugation. Tissue pellets were resuspended in M-199-H supplemented with sucrose 0.5 M (Sigma-Aldrich), EGTA 1.8 mM, penicillin/streptomycin 100 µg/mL and BSA 0.2% before incubated 5 min at 37 °C. Following a second centrifugation at $250 \times g$ for 5 min, tissue pellets were resuspended in DMEM/HamF12 medium supplemented with L-glutamine, penicillin/streptomycin and FBS 1%. Tissue samples were dissociated using a plastic dounce

homogenizer by gently pressure. Preparations were centrifuged at $500 \times g$ for 10 min and resulting pellets resuspend in DMEM/HamF12 medium supplemented with L-glutamine 2 mM (Invitrogen), penicillin/streptomycin and FBS 5%. Cell suspension was used to seed 12-well plates at a final concentration of $3 \times 10^5$ cells per well and were grown at 37 °C in a humidified air 5% $CO_2$ incubator. Granulosa cell isolation protocol was adapted from previously described method (Campbell, 1979).

## COC isolation and staining

Following hyperstimulation protocol described above. Cumulus-oocytes complexes were collected by oviduct dissection. COC were incubated in PBS1X-PFA 4% for 30 min at room temperature by gently rocking. After multiple washes in PBS1X-BSA 0.1% 3, COC were permeabilized using PBS1X-Tween 0.5%/Triton 0.1% solution during 1 h at room temperature by gently rocking. Then, samples were washed in PBS-BSA 0.1% and incubated in a blocking solution PBS1X-Tween 0.5%, Goat serum 1% BSA 0.1% Triton 0.1% 1 h at room temperature by gently rocking. Preparation was incubated with Anti CD45 (BD Pharmingen 550539) in PBS-BSA 0.1% overnight at 4 °C by gently rocking. Then, they were incubated with goat Anti Rat Alexa 568 (Invitrogen) in PBS-BSA 0.1% overnight at 4 °C by gently rocking and stained with Hoescht (Invitrogen) 1/10,000 in PBS-BSA 0.1% 30 min at room temperature by gently rocking. Samples were then analyzed with Zeiss AxioImager station with Apotome2 and Zeiss 800 Airyscan for 3D confocal imaging. Pictures analysis were conducted using Imaris software v10.0.1 (Oxford Instruments).

## Bone marrow transplantation

Bone marrows were collected from all genotypes. Briefly, femurs were collected and flushed using syringes filled with PBS1X and mounted with 25 G needles by punction of the medullar cavity. Bone marrow cells were then centrifuged at $500 \times g$ for 10 min and resuspend to obtain a $8 \times 10^6$ cells/mL suspension. Then, 200 μL of the cell suspension was injected to NSG mice through the tail vein route. Blood samples were collected closed to 1-month post-transplantation to validate grafting success by flow cytometry using red blood lysis buffer according manufacturer instructions (Abcam, Ab204733). After blocking with anti-CD16/CD32 (Pharmigen 553141), samples were analyzed using Live/Dead fixable green dead cell stain kit (Invitrogen L34969) and anti-CD19 APC clone 6D5 (Biolegend 115512) or isotype control clone K2758 (Biolegend 400512). Cytometry analyses were performed using Attune Nxt station (Invitrogen). One month after transplantation, NSG-implanted mice were stimulated according to hormonal injection procedure indicated above. Ovaries were collected and analyzed by immunohistochemistry for CD45 and MHC-II detection.

## RNA sequencing and dataset analysis

RNA sequencing was performed by GenomEast platform (France Genomique Consortium, ANR-10-INBS-009). Libraries were performed using the TrueSeq Stranded mRNA Library Prep kit (Illumina, San Diego, USA). Libraries quality and quantification were measured by capillarity electrophoresis and sequenced using Illumina HiSeq 4000. Reads were cleaned and filtered using cutadapt v3.2 (Martin, 2011) and FASTQC v0.11.7 (https://www.bioinformatics.babraham.ac.uk/projects/fastqc/). Reads were aligned using mouse reference genome mm10 or human reference genome hg38 with Hisat2 v2.2.1 (Kim et al, 2019). Reads counts were reported for each annotated genes using R and convert into RPKM. Datasets are available using accession numbers: GSE22134, GSE222135, GSE271363. Principal component analysis were generated for each datasets using plotPCA function from R package Deseq2 (Love, 2014) and common gene signature were explored with DeepVenn (Hulsen, 2022). Heatmaps were generated with pheatmap package in R. Representation is based on median centered RPKM levels and genes organized by unsupervised hierarchical clustering. Gene ontology analyses were conducted using cluster profiler package in R (Wu et al, 2021; clusterProfiler: an R Package for Comparing Biological Themes Among Gene Clusters | OMICS: A Journal of Integrative Biology). Gene Set Enrichment Analyses were also done using "gseGO" method from Cluster Profiler R package (Wu et al, 2021; clusterProfiler: an R Package for Comparing Biological Themes Among Gene Clusters | OMICS: A Journal of Integrative Biology). Immune infiltrate analysis was investigated by deconvolution method with CIBERSORTx (https://cibersortx.stanford.edu) (Newman et al, 2019). NF-κB gene list was obtained from the Gilmore lab ressourses (https://www.bu.edu/nf-kb/gene-resources/target-genes/). Gene expression datasets were analyzed using GSEA 4.1.0 with genesets either from the MSigDB Hallmark gene set or from custom-curated genesets. Granulosa signature was identified using the list of the dataset GSE140371 (Madogwe et al, 2020). Then, the leading-edge gene lists commonly deregulated in the WTvsDKO and TG-AMHβvsDKO differential were used to identify PMSG responding clusters. Single-cell sequencing were used to analyzed both NR1H2 and TXNIP in granulosa cell from growing follicle using GSE107746 dataset (Zhang et al, 2018). Boxplots were generated using R package ggplot2. Single-cell sequencing analysis from adult ovary were visualized by tSNE representation from GSE118127 dataset using the Ovogrowth server (http://ovogrowth.net/) (Fan et al, 2019). Single-cell analysis conducted on mouse PCOS model GSE268919 (Luo et al, 2024) were performed using Seurat (Version 4.4.0). UMAP visualization was conducted across samples to assess batch effects. Correction has been made using Harmony package (Version 1.2.3) (Korsunsky et al, 2019). Visualizations with UMAP, blox plot, violin plot were conducted using Seurat (Version 4.4.0), Seurat Extend (Version 1.1.4) (Hua et al, 2024), and ggplot2 (Version 3.5.1) (Wickham, 2016).

## Patient cohort

Written informed consent was obtained for inclusion of the cumulus cells from 16 patients (5 women with PCO and 11 control patients without dysovulation in the Germetheque biobank with the approval of the local committee (Committee for Personal Protection DC 2008 558 nimner: 20200703, Trial registration number: NCT04715858) in Clermont university hospital. Informed consent was obtained from all patients and confirm that the experiments conformed to the principles set out in the WMA Declaration of Helsinki and the Department of Health and Human Services Belmont Report. Clinical features are summarized in Dataset EV5.

**The paper explained**

**Problem**

Assisted reproductive technology is a broadly used practice worldwide. However, some patients present risks of ovarian hyperstimulation in response to hormonal treatments. This is particularly the case for patients suffering from polycystic ovary syndrome. The molecular etiology of this increased sensitivity to hormonal stimulation remains unclear and significant biomarkers predictive of this response are lacking.

**Results**

Here, we reveal the central role that Liver X Receptors play in controlling the response to hormonal treatment during ovary stimulation. Thus, the hormonal protocol activates the inflammation pathways which are necessary for the natural process of ovulation. In parallel, they activate the Liver X Receptor pathway to curb this inflammation and ensure a proportionate response. These data obtained from a mouse model are faithfully found in patients with PCOS unmasking an unexpected immune infiltrate within cumulus-oocyte complex.

**Impact**

Our results made it possible to identify a signature of ovarian hyperstimulation associated with the pathological response. The presence of the immune infiltrate in the cumulus-oocyte complex suggests that the serum dosage of certain proinflammatory cytokines may constitute a significant biomarker for monitoring the ovarian response to hormonal treatment.

## Statistical analyses

The number of individual samples are indicated in figures legends. Statistical analyses were conducted with GraphPad Prism v9. Statistical analyses were performed using the two-tailed Mann–Whitney test, Ordinary one-way ANOVA test and Kolmogorov–Smirnov test. All bars represent means ± SD. Significance is reported according to $*P < 0.05$, $**P < 0.01$, $***P < 0.001$ and $***P < 0.0001$. Exact $P$ values for these statistical comparisons are shown in Appendix Table S1.

## Graphics

Icons and graphical supports have been created with BioRender.com. Agreement number: GI26YPY0PW.

## Data availability

The datasets produced and/or used in this study are available in the following databases: RNAseq dataset: Transcriptomic analysis of corona radiata cells after hormonal stimulation (PMSG/hCG) GSE222134, RNAseq dataset: Transcriptomic analysis of mouse ovary post-PMSG induction 40 h GSE222135, RNAseq dataset: Sequencing mRNA of human cumulus cells obtained from patients with PCOS or CONTROL GSE271363, single-cell RNAseq dataset: Integrated Single-Cell and Spatial Transcriptomics Reveal Androgen-Driven Disruptions in PCOS Ovarian Microenvironment GSE268919.

The source data of this paper are collected in the following database record: biostudies:S-SCDT-10_1038-S44321-025-00251-1.

## Peer review information

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

## Acknowledgements

We thank S. Plantade, K. Ouchen, P. Mazuel, and E. Dubosclard for animal housing and care. KGN cells were kindly provided by Dr. M. Pannetier (INRAe, France) and Dr. J. Cocquet (Institut Cochin, France). We thank Dr. R. Guiton for critical advice regarding immune infiltration analysis as well as cytometry experiments. We thank D. Garcia Garcia for help regarding PCOS dataset analysis. Histological and immunohistochemistry have been performed using the Anip@th platform (https://www.igred.fr/en/plateform/anipath-histopathologie/), microscopy and imaging analyses using the CLIC platform (https://www.igred.fr/en/plateform/clic/) and cytometry using the SC3 platform (https://www.igred.fr/en/plateform/sc3-single-cell-cell-culture-facilities/). We warmly thank K. Gates for careful editing of the manuscript.

## Author contributions

**Sarah Dallel:** Conceptualization; Data curation; Formal analysis; Investigation; Methodology; Writing—original draft; Writing—review and editing. **Manon Despalles:** Formal analysis; Investigation. **Margaux Tore:** Investigation. **Yoan Renaud:** Software; Investigation; Bioinformatic work. **Ayhan Kocer:** Investigation; Methodology. **Christelle Damon-Soubeyrand:** Investigation; Visualization. **Pierre Pouchin:** Visualization. **Caroline Vachias:** Visualization. **Katia Boutourlinsky:** Investigation; Methodology. **Céline Gonthier-Gueret:** Investigation; Methodology. **Angélique De Haze:** Investigation; Methodology. **Phelipe Sanchez:** Investigation; Methodology. **Jean-Christophe Pointud:** Investigation; Methodology. **Erwan Bouchareb:** Formal analysis; Investigation. **Marine Vialat:** Investigation. **Aurélie Lagarde:** Investigation. **Cristina Gulunga:** Resources; Methodology. **Laure Chaput:** Resources; Methodology. **Aurelie Vega:** Resources; Investigation; Methodology. **Florence Brugnon:** Conceptualization; Resources. **Igor Tauveron:** Resources. **Amalia Trousson:** Resources; Data curation; Software; Investigation; Methodology. **Cyrille de Joussineau:** Investigation; Writing—review and editing. **Françoise Degoul:**

Writing—original draft. **Laurent Morel**: Writing—original draft. **Jean Marc Lobaccaro**: Conceptualization; Writing—original draft; Project administration; Writing—review and editing. **Salwan Maqdasy**: Conceptualization; Investigation; Methodology; Writing—original draft; Project administration. **Silvère Baron**: Conceptualization; Resources; Data curation; Software; Formal analysis; Supervision; Validation; Investigation; Visualization; Methodology; Writing—original draft; Project administration; Writing—review and editing.

Source data underlying figure panels in this paper may have individual authorship assigned. Where available, figure panel/source data authorship is listed in the following database record: biostudies:S-SCDT-10_1038-S44321-025-00251-1.

## Disclosure and competing interests statement

The authors declare no competing interests.

