## [Peer Review File · EMBO Molecular Medicine]

LXR Pathway Drives Hormonal Response Intensity in PolyCystic Ovary Syndrome.

Sarah Dallel, Manon Despalles, Margaux Tore, Yoan Renaud, Ayhan Kocer, Christelle Damon-Soubeyrand, Pierre Pouchin, Caroline Vachias, Katia Boutourlinsky, Céline Gonthier-Gueret, Angélique De Haze, Phelipe Sanchez, Jean-Christophe Pointud, Erwan Bouchareb, Marine Vialat, Aurélie Lagarde, Cristina Gulunga, Laure Chaput, Aurélie Vega, Florence Brugnon, Igor Tauveron, Amalia Trousson, Cyrille de Joussineau, Françoise Degoul, Laurent Morel, Jean Marc Lobaccaro, Salwan Maqdasy, Silvère Baron

Corresponding author: Silvère Baron (silvere.baron@uca.fr)

Review Timeline:

Submission Date:	16th Jul 24
Editorial Decision:	18th Jul 24
Appeal:	18th Jul 24
Editorial Decision:	28th Aug 24
Revision Received:	18th Dec 24
Editorial Decision:	9th Jan 25
Appeal:	14th Jan 25
Editorial Decision:	17th Jan 25
Revision Received:	4th Apr 25
Editorial Decision:	15th Apr 25
Revision Received:	24th Apr 25
Accepted:	29th Apr 25

Editor: Zeljko Durdevic

Transaction Report:

18th Jul 2024

Decision on your manuscript EMM-2024-20298

Dear Dr. Baron,

Thank you for submitting your manuscript to EMBO Molecular Medicine. Upon receipt, manuscripts can sometimes be evaluated by the Scientific Editors to deal in a timely fashion with a large number of submissions. In this case, I am afraid that we concluded that your manuscript is not well suited for publication in EMBO Molecular Medicine and have therefore decided not to proceed with peer review.

While potentially of interest to the more immediate community, I am afraid that due to its nature, the article doesn't fit well within EMBO Molecular Medicine as we focus primarily on these studies that provide functional novel insights of clinical and/or translational significance, but also that are conceptually novel and of broad interest. As we do not feel that this is the case here, we therefore cannot offer further consideration to your manuscript.

I am sorry to have to disappoint you on this occasion; in the interest of time, I am providing you with an early decision that will allow you to submit your manuscript elsewhere without any further delays.

Please rest assured that this is not a judgment of the quality or interest of your work but a decision based on appropriateness for EMBO Molecular Medicine.

Yours sincerely,

Zeljko Durdevic

As a service to authors, EMBO provides authors with the possibility to transfer a manuscript that one journal cannot offer to publish to another EMBO publication. The full manuscript and if applicable, reviewers reports are automatically sent to the receiving journal to allow for fast handling and a prompt decision on your manuscript. For more details of this service, and to transfer your manuscript to another EMBO title please click on Link Not Available

Dear Dr. Zeljko Durdevic,

First, I really want to thank you for this quick response regarding the submission of this article (EMM-2024-20298). I carefully pay attention of the elements which guided your choice not to retain this work for revision. However, being an avid reader of the EMBO molecular medicine journal, and occasional reviewer, I am convinced that the content of this article completely corresponds to your editorial line of your journal. I would therefore like to draw your attention to two points:

- The **impact in translational medicine** of the proposed work, with the **identification of a molecular signature in patients with PCOS in those the etiology of an ovarian hyper stimulation remain completely unknown.**

- The **broad audience of interest** represented by the results obtained taking into account the **number of patients included in medically assisted reproduction protocols worldwide.**

Therefore, I request from you a re-evaluation of the potential of this article for publication in your journal. I remain convinced that, by reading these few arguments, you will be able to **give this work the chance to be peer-reviewed**. I remain at your disposal for any information that could enlighten your decision.

Sincerely

Dr. Silvere Baron

28th Aug 2024

Dear Dr. Baron,

Thank you for the submission of your manuscript to EMBO Molecular Medicine and please accept my apologies for the delay in getting back to you due to the holiday season. We have now received feedback from the two reviewers who agreed to evaluate your manuscript. Both referees recognize interest of the study but also raise important concerns that should be addressed in a major revision. Particular attention should be given to confirmation of the main findings in PCOS patient cells. If you would like to discuss further the points raised by the referees, I am available to do so via email or video. Let me know if you are interested in this option.

We would welcome the submission of a revised version within three months for further consideration. Please let us know if you require longer to complete the revision.

I look forward to receiving your revised manuscript.

Yours sincerely,

Zeljko Durdevic

We require:

- 1) A .docx formatted version of the manuscript text (including legends for main figures, EV figures and tables). Please make sure that the changes are highlighted to be clearly visible.
- 2) Individual production quality figure files as .eps, .tif, .jpg (one file per figure). For guidance, download the 'Figure Guide PDF': (<https://www.embopress.org/page/journal/17574684/authorguide#figureformat>).
- 3) A .docx formatted letter INCLUDING the reviewers' reports and your detailed point-by-point responses to their comments. As part of the EMBO Press transparent editorial process, the point-by-point response is part of the Review Process File (RPF), which will be published alongside your paper.
- 4) A complete author checklist, which you can download from our author guidelines (<https://www.embopress.org/page/journal/17574684/authorguide#submissionofrevisions>). Please insert information in the checklist that is also reflected in the manuscript. The completed author checklist will also be part of the RPF.
- 5) Please note that all corresponding authors are required to supply an ORCID ID for their name upon submission of a revised manuscript.

6) It is mandatory to include a 'Data Availability' section after the Materials and Methods. Before submitting your revision, primary datasets produced in this study need to be deposited in an appropriate public database, and the accession numbers and database listed under 'Data Availability'. Please remember to provide a reviewer password if the datasets are not yet public (see <https://www.embopress.org/page/journal/17574684/authorguide#dataavailability>).

13) Author contributions: You will be asked to provide CRediT (Contributor Role Taxonomy) terms in the submission system. These replace a narrative author contribution section in the manuscript.

14) A Conflict of Interest statement should be provided in the main text.

16) Include a Reagents and Tools Table as part of the Methods section, which can be downloaded from our author guidelines (<https://www.embopress.org/page/journal/17574684/authorguide#structuredmethods>)

**** Reviewer's comments ****

Referee #1 (Comments on Novelty/Model System for Author):

The work is mainly done in rodent models and confirmation in patients with PCOS is needed before any model can be proposed in humans.

Referee #1 (Remarks for Author):

The authors study the mechanism of ovarian hyperstimulation (OHSS) in response to hormonal treatments observed in assisted reproductive technology (ART) especially in patients suffering from polycystic ovary syndrome (PCOS). This is an important question to address and the identification of biomarkers might be useful to predict OHSS in non-PCOS patients undergoing ART. The authors had previously observed that LXR DKO mice exhibit an OHSS-like phenotype in response to ovarian stimulation suggesting a protective role of LXR against hyperstimulation. They show that the LXR agonist GW3965 can prevent OHSS in WT mice and confirm the central role of LXR in this process. They show that LXR β -dependent regulation is important in granulosa cells using knock-out models, rescued in a TgAMH_Lxr β mice.

Using the same models they show that hormonal hyperstimulation is associated with immune cell infiltration in the absence of LXR in LXR DKO mice, due to impaired granulosa cell hormonal response.

The authors used RNA sequencing data from ovaries of WT, LXR DKO and TgAMH-LXR β mice they show that hormonal stimulation is associated with granulosa cell-dependent inflammation with increased expression of genes of the NF- κ B pathway such as Nlrp3 in the absence of LXR. They further identify the most deregulated gene Txnip gene encoding the Thioredoxin Binding Protein (TXNIP), which expression increase in the LXR DKO ovaries. They concluded that TXNIP is a possible candidate to support OHSS phenotype.

The authors studied the expression of other genes encoding the inflammasome components and observe that all of them were upregulated in LXR DKO mice compared to WT and TgAMH-LXR β mice. Treatment of mice with an NLRP3 inhibitor MCC950 yielded a significant decrease in the hemorrhagic foci in LXR DKO mice.

Finally, they use human cumulus granulosa RNAseq datasets from controls and PCOS patients and compared them to rodent RNAseqs. They identified a hyperstimulation gene signature that aggregate 96 genes. Using single cell RNA seq they show that that a number of genes were expressed in the immune cell compartment. Among them they identified TNFRSF1B and ILB already identified in LXR DKO mice ovaries.

The authors concluded that TXNIP is a key player in the integration of hormonal signals and in the inflammatory status of the ovary and that Txnip is a strong candidate to be controlled by the NF- κ B pathway. They propose that targeting of LXR during hormonal stimulation in ART especially in PCOS patients could be a relevant therapeutic pathway for patients at risk of OHSS.

This work is interesting and propose a new pathophysiology of OHSS.

However, the work is mainly done in rodent models and the confirmation in humans needs to be studied in more depth to show how the candidates identified here can be used as biomarkers of OHSS in humans, as proposed by the authors.

1- There is very little information on PCOS patients. Information on recruitment, their origin, complete clinical information together with hormonal assays and ultrasound data should be provided, as well as for controls (FSH LH androgens especially, US). A table can be provided.

2- Granulosa cells and immune cells from PCOS patients should be used to confirm the proposed model. QPCR of the proposed biomarkers and immunocytochemical studies should be performed. Treatment of cultured PCOS cells with LXR agonist, NF- κ B inhibitor, TXNIP SiRNAs is necessary to confirm the model proposed, mainly based on studies in rodent models and a comparison of RNA sequencing between human and mouse granulosa cells.

3- A role for MIR1224 is proposed. QPCR assays might also be performed in human granulosa cells from PCOS patients studied

here to support the role proposed.

Referee #2 (Comments on Novelty/Model System for Author):

This paper is well-organized and an interesting topic for the readers of reproductive endocrinology. It provides the new insight in the role of granulosa cells in inflammatory abnormalities during the ovulation process. However, there are several points that need to be revised.

Referee #2 (Remarks for Author):

This paper revealed the pathological mechanism of ovarian hyperstimulation. Firstly, Authors showed the role of Liver X receptors via hormonal stimulation by using Knock out mouse models. Authors also reveal the molecular mechanism of ovarian hyperstimulation focusing on inflammatory pathways by RNA-Seq of ovary and granulosa cells of Knockout mouse models, and granulosa cells in PCOS patients. This paper is interesting for readers regarding reproductive medicines and provides the new insight in the role of granulosa cells in inflammatory abnormalities during the ovulation process. However, the link between PCOS and dysregulation of inflammatory driven by LXR is weak. Some revisions are required to be published. Below are some suggestions for improvement.

Major Revisions:

1. OHSS phenotypes are defined as the presence of hemorrhagic cysts. What is the definition of hemorrhagic foci? VEGF production and Vascular permeability need to be examined in Figure1. It is unclear whether OHSS phenotype is truly rescued in TGAMH-LKRB models.
2. In Figure 1F, the number of normal oocyte is decreased in knockout mouse. How about the ovulation rate (the total number of oocytes)? The number of oocytes retrieved is crucial for OHSS phenotypes.
3. In Figure2, why did you perform RNA-seq using 40h post PMSG injections? In Figure2B, the number of hemorrhagic foci is affected the most in 46 h post PMSG. Please explain the validity.
4. Which is crucial for infiltration of immune response via hormonal stimulation, mural granulosa cells or cumulus cells? In figure3D, p65 was localized in mural granulosa cells. How about in cumulus cells?
5. As I mentioned in No.3, why did you use post 48hrs granulosa cells in Figure4? If abnormal inflammatory cascade is triggered by granulosa cells, its function needs to be examined prior to the activation of inflammation.
6. The expression of LXR and TXNIP in granulosa cells need to be shown in RNA-Seq of PCOS patients compared with control patients. It is unclear whether OHSS in PCOS patients are induced by LXR dependent inflammation.

Minor revisions

1. Please spell out all the abbreviations. Like PPAR, AP1, NF-kB, NLRP3, ASC, CASP1.

Referee #1 (Comments on Novelty/Model System for Author):

The work is mainly done in rodent models and confirmation in patients with PCOS is needed before any model can be proposed in humans.

Referee #1 (Remarks for Author): **Editing has been highlighted in yellow**

The authors study the mechanism of ovarian hyperstimulation (OHSS) in response to hormonal treatments observed in assisted reproductive technology (ART) especially in patients suffering from polycystic ovary syndrome (PCOS). This is an important question to address and the identification of biomarkers might be useful to predict OHSS in non-PCOS patients undergoing ART.

The authors had previously observed that LXR DKO mice exhibit an OHSS-like phenotype in response to ovarian stimulation suggesting a protective role of LXR against hyperstimulation. They show that the LXR agonist GW3965 can prevent OHSS in WT mice and confirm the central role of LXR in this process. They show that LXR β -dependent regulation is important in granulosa cells using knock-out models, rescued in a TgAMH_Lxr β mice.

Using the same models they show that hormonal hyperstimulation is associated with immune cell infiltration in the absence of LXR in LXR DKO mice, due to impaired granulosa cell hormonal response.

The authors used RNA sequencing data from ovaries of WT, LXR DKO and TgAMH-LXR β mice they show that hormonal stimulation is associated with granulosa cell-dependent inflammation with increased expression of genes of the NF- κ B pathway such as Nlrp3 in the absence of LXR. They further identify the most deregulated gene Txnip gene encoding the Thioredoxin Binding Protein (TXNIP), which expression increase in the LXR DKO ovaries. They concluded that TXNIP is a possible candidate to support OHSS phenotype.

The authors studied the expression of other genes encoding the inflammasome components and observe that all of them were upregulated in LXR DKO mice compared to WT and TgAMH-LXR β mice. Treatment of mice with an NLRP3 inhibitor MCC950 yielded a significant decrease in the hemorrhagic foci in LXR DKO mice.

Finally, they use human cumulus granulosa RNAseq datasets from controls and PCOS patients and compared them to rodent RNAseqs. They identified a hyperstimulation gene signature that aggregate 96 genes. Using single cell RNA seq they show that that a number of genes were expressed in the immune cell compartment.

Among them they identified TNFRSF1B and ILB already identified in LXR DKO mice ovaries.

The authors concluded that TXNIP is a key player in the integration of hormonal signals and in the inflammatory status of the ovary and that Txnip is a strong candidate to be controlled by the NF- κ B pathway. They propose that targeting of LXR during hormonal stimulation in ART especially in PCOS patients could be a relevant therapeutic pathway for patients at risk of OHSS.

This work is interesting and propose a new pathophysiology of OHSS.

However, the work is mainly done in rodent models and the confirmation in humans needs to be studied in more depth to show how the candidates identified here can be used as biomarkers of OHSS in humans, as proposed by the authors.

1- There is very little information on PCOS patients. Information on recruitment, their origin, complete clinical information together with hormonal assays and ultrasound data should be provided, as well as for controls (FSH LH androgens especially, US). A table can be provided.

*** We agree with Reviewer#1 remarks. Given this cohort is retrospective we only get a partial answer to these requests. We hope that these additions give a sufficient overview of informations regarding the patient cohort.

Table S5 has been edited to include a maximum of informations.

Recruitment: monocentric Clermont-Ferrand hospital

Origin: French ethical rules prohibit us from giving these informations

Hormonal assays: indicated when available

Ultrasound data: Antral follicle count (AFC) have been added with menstrual cycles problems

2- Granulosa cells and immune cells from PCOS patients should be used to confirm the proposed model. QPCR of the proposed biomarkers and immunocytochemical studies should be performed.

*** We totally agree with Reviewer#1. According to these recommendations, we have confirmed 9 targets by RT-qPCR. Unfortunately, we do not have samples for immunocytochemical studies. We have modified Figure 7G and added Figure S9C to include these new set of results. We hope that with adding comforts enough results from PCOS patients.

Treatment of cultured PCOS cells with LXR agonist, NF- κ B inhibitor, TXNIP SiRNAs is necessary to confirm the model proposed, mainly based on studies in rodent models and a comparison of RNA sequencing between human and mouse granulosa cells.

*** According to Reviewer#1 recommendations, we have explored cultured granulosa cell in a context of PCOS. We used KGN cell line that has been reported to mimic PCOS-related cell injury when exposed to TNF α (Li et al. Bioengineered 2021, PMID: 34637688). As presented below, following Reviewer#1 suggestions, we have monitored TXNIP accumulation in response to TNF α or TNF α combined with GW3965 (LXR agonist) (Figure R1A). We did not observe any changes in TXNIP accumulation, whatever the antibody used. Next, to phenocopy transgenic mouse models findings, we have engineered KGN-CRISPR-Cas9 model to delete both LXR α et LXR β .

We have successfully obtained LXR depleted cells given the loss of stimulation of the ABCG1 canonical target in response to GW3965 (Figure R1B-C). In these cell lines, we have examined *TXNIP* as well as inflammasome components such as *NLRP3*, *PYCARD*, *CASP1* by RT-qPCR and western blot (Figure R1B-C). First, we confirm TNF α efficiency that led to *NLRP3*, *PYCARD*, *CASP1* and *IL1B* expression stimulation in KGN-Cas9GFP. Nevertheless, we have not observed any change in *TXNIP* expression (Figure R1B). In terms of relative expression, LXR ablation in CRISPR cell lines deeply suppressed expression of *NLRP3* *PYCARD* and *IL1B* without affecting *CASP1* (Figure 1B). Regarding *TXNIP* expression, LXR invalidation decreases its expression in absence of TNF α but remains equivalent for all the other conditions (Figure R1B). These finding was confirmed by western blot (Figure R1C). Finally, qPCR and western blots for inflammasome components give really heterogenous results making any extrapolation with mice models hazardous. Given these results have revealed an absence of *TXNIP* deregulation in the condition used, despite already published data, we have not pushed further with *TXNIP* knockdown or NF- κ B inhibitors. We have no rational interpretation to explain why we have failed to reproduce previous work reporting TNF α -induced expression of *TXNIP* (Cell line drifting, source of reagents...). Altogether, we conclude that, to go deeper in the mechanism, we need to screen other human granulosa cell lines to find a robust cellular model of PCOS. Unfortunately, we do not have such model for the article.

Figure R1 : A - Accumulation of ABCG1, TXNIP and ACTIN protein in KGN granulosa cells exposed to TNF 100ng/mL with or without GW3965 1 μ M, TXNIP has been detected using antibody ab188865 from Abcam or antibody NBP1-54578 from NOVUS Biotechnology. B - Accumulation of ABCG1, TXNIP, NLRP3, ASC/PYCARD and ACTIN protein in KGN-Cas9GFP or KGN Cas9GFP-CRISPR-LXR $\alpha\beta$ cells exposed to TNF 100ng/mL with or without GW3965 1 μ M. C - Expression analysis by RT-qPCR of *NLRP3*, *ABCG1*, *PYCARD*, *TXNIP*, *IL1B* and *CASP1* genes in KGN-Cas9GFP or KGN Cas9GFP-CRISPR-LXR $\alpha\beta$ cells exposed to TNF 100ng/mL with or without GW3965 1 μ M.

3- A role for MIR1224 is proposed. QPCR assays might also be performed in human granulosa cells from PCOS patients studied here to support the role proposed.

*** Following Reviewer#1 suggestion, we have tried to detect mir1224p by RT-qPCR using either human patient samples or transgenic mouse samples. We have not detected any amplification of this microRNA. Thus, we have decided to remove this part of the discussion considering these results.

Referee #2 (Comments on Novelty/Model System for Author): **Editing has been highlighted in blue**

This paper is well-organized and an interesting topic for the readers of reproductive endocrinology. It provides the new insight in the role of granulosa cells in inflammatory abnormalities during the ovulation process. However, there are several points that need to be revised.

Referee #2 (Remarks for Author):

This paper revealed the pathological mechanism of ovarian hyperstimulation. Firstly, Authors showed the role of Liver X receptors via hormonal stimulation by using Knock out mouse models. Authors also reveal the molecular mechanism of ovarian hyperstimulation focusing on inflammatory pathways by RNA-Seq of ovary and granulosa cells of Knockout mouse models, and granulosa cells in PCOS patients. This paper is interesting for readers regarding reproductive medicines and provides the new insight in the role of granulosa cells in inflammatory abnormalities during the ovulation process. However, the link between PCOS and dysregulation of inflammatory driven by LXR is weak. Some revisions are required to be published. Below are some suggestions for improvement.

Major Revisions:

1. OHSS phenotypes are defined as the presence of hemorrhagic cysts. What is the definition of hemorrhagic foci? VEGF production and Vascular permeability need to be examined in Figure 1.

It is unclear whether OHSS phenotype is truly rescued in TGAMH-LKrb models.

*** Reviewer#2 comment is relevant. We mentioned hemorrhagic "foci" but we agree that the correct denomination is hemorrhagic cyst. We have thus edited the manuscript accordingly. Indeed, VEGF and vascular permeability are hallmarks of OHSS. In consequence, we have performed some RT-qPCR using mouse ovary samples. We have not monitored significant modification in VEGF expression. We have tested two various antibodies against VEGF whose did not give relevant immunostaining. We have performed endomucin detection (endothelial cell marker) to appreciate vascular architecture (Figure S3A) and obtained significant results. We have added vascular permeability measurement that confirm an increased permeability in LXR DKO whatever the LXRb granulosa rescue as well as endomucin staining. These results have been inserted in Figure S3A.

2. In Figure 1F, the number of normal oocyte is decreased in knockout mouse. How about the ovulation rate (the total number of oocytes)? The number of oocytes retrieved is crucial for OHSS phenotypes.

*** We thank Reviewer#2 for this remark and apologize for this oversight. Total number of oocytes has been added in Figure 1 and count of normal oocytes move in Figure S2.

3. In Figure 2, why did you perform RNA-seq using 40h post PMSG injections? In Figure 2B, the number of hemorrhagic foci is affected the most in 46 h post PMSG. Please explain the validity.

*** Reviewer#2 has noticed that hemorrhagic cyst number reach the maximum at late-stage post PMSG (50-66h). We have chosen to investigate transcriptomic profile at 40h post PMSG to get a better chance to identify molecular mechanism involved. At this time point, we have already observed significant phenotype occurrence of hemorrhagic cysts and place analysis before ovulation that could complexified transcriptomic signatures given granulosa cells undergo luteinization.

4. Which is crucial for infiltration of immune response via hormonal stimulation, mural granulosa cells or cumulus cells? In figure 3D, p65 was localized in mural granulosa cells. How about in cumulus cells?

*** Reviewer#2 question regarding p65 localization in granulosa compartment is of interest. As requested, we have checked for p65 localization in cumulus cells and added the results in Figure S5. We did not observe any modification of p65 delocalization in the nucleus whatever the genotype indicating that deregulation of the NF-kB pathway would be restricted to mural granulosa cells. To go further, we have analyzed TXNIP between mural and cumulus cells leading to conclude that TXNIP deregulation is restricted to mural granulosa.

5. As I mentioned in No.3, why did you use post 48hrs granulosa cells in Figure 4?

*** The reason we have used 48h post stimulation came from that we explored a dataset already published by Madogwe et al. (ref #21) as a support to identify specific granulosa cell signature. Unfortunately, this group did not investigate earlier time points.

If abnormal inflammatory cascade is triggered by granulosa cells, its function needs to be examined prior to the activation of inflammation.

*** We would like to thank Reviewer#2 for this comment. To investigate shorter time point could lead to identify molecular pathway involved. This is the reason why we focused our attention on the 40h post PMSG time point. To go further and investigate that TXNIP could be deregulated in early stages, we have performed immunodetection on ovary of various genotype 24h post PMSG (Figure S7). We have not observed any modification of the staining at this time point indicating that lack of LXR in ovary plays crucial spatial (Mural granulosa) and temporal (between 24 to 40h post PMSG) roles during folliculogenesis.

6. The expression of LXR and TXNIP in granulosa cells need to be shown in RNA-Seq of PCOS patients

compared with control patients. It is unclear whether OHSS in PCOS patients are induced by LXR dependent inflammation.

*** We thank Reviewer#2 for this question that give us the opportunity to analyze deeper our datasets and phenotypes. Mouse models indicate that LXR play an important role of inflammasome regulation. Thus, we have compared analysis expression of genes encoding inflammasome component between our three datasets: total ovary 40h post PSMG, mouse cumulus granulosa cells and woman cohort cumulus granulosa cells. Interestingly, immune signature is present in all datasets while inflammasome signature is only present in the total ovary dataset (Figure S9). This observation makes sense regarding TXNIP localization between mural and cumulus granulosa cells (Figure 5). We conclude that inflammation in LXR DKO mouse ovary initiates mainly from mural cell dysfunction leading to attract immune infiltration into the follicle including cumulus compartment. It is not possible to conclude that PCOS patients present a LXR dysregulation since we do not have access to mural cell samples from these patients, but we have observed that LXR-null model share many common features with the human syndrome.

Minor revisions

1. Please spell out all the abbreviations. Like PPAR, AP1, NF-kB, NLRP3, ASC, CASP1.

***We apologize for this oversight. Abbreviations have been edited.

To review GEO accession GSE271363:

Go to <https://www.ncbi.nlm.nih.gov/geo/query/acc.cgi?acc=GSE271363>

Enter token uderguqgfjgjjin into the box

9th Jan 2025

Decision on your manuscript EMM-2024-20298-V3

Dear Dr. Baron,

Thank you for the submission of your manuscript to EMBO Molecular Medicine. We have now received feedback from the two reviewers who agreed to re-evaluate your manuscript. As you will see from their reports pasted below, while the referees #2 supports publication of the manuscript, referee #1 acknowledges the addition of new information but he/she remains critical particularly regarding, the lack of evidence that the model proposed in rodent species is relevant for humans.

From our side, we appreciate additional work done to address referees' criticism and we do recognize the potential interest of your findings; however, we agree with the referee and are not persuaded that your manuscript provides the sort of translational implication we would expect in an EMBO Molecular Medicine article. Therefore, I am afraid that we cannot offer to consider the manuscript further. At this point, we recommend, to avoid further frustration, submission of the manuscript to any other of the large number of reputable journals in the area.

I understand that this is disappointing and regret that I could not bring better news this time. Please rest assured that this is not a judgment of the quality or interest of your work, but a decision based on appropriateness for EMBO Molecular Medicine. I wish you luck in identifying a better suited journal for the publication of your study.

Yours sincerely,

Zeljko Durdevic

***** Reviewer's comments *****

Referee #1 (Remarks for Author):

In the revised version of the manuscript Dallel S et al provide new information.

1. Regarding information on PCOS patients, the authors added details in Table S5, including available hormonal assays and ultrasound data, as well as information on patient recruitment at the Clermont-Ferrand hospital. To confirm the proposed model, the authors performed RT-qPCR on 9 targets and modified Figures 7G and S9C to include these results. They were unable to conduct immunocytochemical studies due to a lack of samples.
2. The authors explored granulosa cells in the context of PCOS using the KGN cell line and examined the accumulation of TXNIP and other inflammasome components. They observed heterogeneous results and did not find any regulation of TXNIP, leading them to conclude that they need to explore other human granulosa cell lines. It would be better to use primary culture of granulosa cells of PCOS patients.
3. Regarding the proposed role of MIR1224, the authors attempted to detect this microRNA by RT-qPCR but were unsuccessful, and thus decided to remove this part of the discussion.

In conclusion the model proposed in rodent species is not confirmed in humans.

Referee #2 (Remarks for Author):

Thank you for providing the revised version of the manuscript. After carefully reviewing the article, I am pleased to confirm that the revisions address the concerns raised in my previous review.
I agree with the revised manuscript.

As a service to authors, EMBO provides authors with the possibility to transfer a manuscript that one journal cannot offer to publish to another EMBO publication. The full manuscript and if applicable, reviewers reports are automatically sent to the receiving journal to allow for fast handling and a prompt decision on your manuscript. For more details of this service, and to transfer your manuscript to another EMBO title please click on Link Not Available

Dear Editor,

I have read your message carefully. First of all, I would like to thank you for taking the time to consider each reviewer's feedback and for providing us with a summary report of this revision. As you mentioned, we have made significant efforts to address the reviewers' concerns. Based on that, we are surprised by your decision not to accept the article. I would appreciate some clarifications regarding Reviewer #1's remarks, which you have cited as grounds for rejecting the article, as well as your concerns about the translational potential raised in your message.

Regarding Reviewer #1's feedback, her/his initial comment was that "*the work is mainly done in rodent models, and the confirmation in humans needs to be studied in more depth to show how the candidates identified here can be used as biomarkers of OHSS in humans, as proposed by the authors.*" Specifically, she/he highlighted that "*Granulosa cells and immune cells from PCOS patients should be used to confirm the proposed model,*" a task we have completed and included in the article as requested.

Subsequently, she/he requested additional work involving siRNA or ligand/inhibitor treatments on patient cells. We attempted to address this by using the KGN cell line, as isolating mural granulosa cells from PCOS patients is virtually unattainable for anyone. Indeed, performing ovariectomy on women with PCOS is exceedingly rare, making it actually impossible to obtain such samples. This request thus appears unrealistic and, more importantly, ethically questionable.

Furthermore, I am having difficulty understanding the rationale behind the remark concerning the lack of clinical translation potential. Hence, in our article, we have explored the implications of our discovery from mouse models on a cohort of PCOS patients and we have observed overlapping markers in human cumulus cells. Therefore, I am uncertain about the journal specific expectations in this regard. As for translational potential, we have proposed monitoring plasma IL1-beta levels to track women during stimulation cycles, as there are currently few, if any, reliable markers for hyperstimulation. Let me directly quote Reviewer #1: "*This work is interesting and proposes a new pathophysiology of OHSS.*"

Finally, I would like to emphasize that Reviewer #1 accepted the paper contingent on major revisions, which we addressed thoroughly by providing well-reasoned responses and adding the necessary results to the article. The implicit rules of peer review necessitate a certain level of fairness, which the editor is expected to uphold. It is thus difficult to comprehend the rejection of the article based on arguments that *per se* should have led reviewer #1 to initially reject the paper rather than to ask request major revisions.

Altogether, I respectfully request that you reconsider the paper and, if necessary, involve the more broadly the editorial board in the review process. I remain fully

available for further discussion via video conference to remove any doubts that may remain regarding the potential of this article.

I hope that the arguments presented in this exchange will help to shift your position.

Best regards,

Silvère BARON

16th Jan 2025

Decision on your manuscript EMM-2024-20298-V4-Q

Dear Dr. Baron,

Thank you for your response to the editorial decision on your manuscript entitled "LXR pathway drives response amplitude to ovarian hormonal stimulation". I have now carefully examined the arguments provided in your letter, the point-by-point response and the referee reports and discussed them with the other members of our editorial team. Additionally, I have sought external advice on the study from an expert in the field.

I regret to inform you that we will not be able to reverse our original decision. We do acknowledge the potential interest of your findings; however, in line with the referee #1 and our editorial assessment the external adviser evaluated the manuscript as not suitable for publication in EMBO Molecular Medicine. The adviser raises the critique particularly regarding, but not limited to, unclear human relevance of the findings as the mice data do not support the human data limiting the translational implications of the study. Therefore, taking in consideration that both our external adviser and referee #1 raise substantial and overlapping concerns I am afraid that we cannot offer further consideration to your article at EMBO Molecular Medicine. At this point, we recommend, to avoid further frustration, submission of the manuscript to any other of the large number of reputable journals in the area.

I understand that this is disappointing and regret that I could not bring better news this time. Please rest assured that this is not a judgment of the quality or interest of your work, but a decision based on appropriateness for EMBO Molecular Medicine. I hope that this negative decision does not prevent you from considering our journal for the publication of your future studies. I wish you luck in identifying a better suited journal for the publication of your study.

Yours sincerely,

Zeljko Durdevic

As a service to authors, EMBO provides authors with the possibility to transfer a manuscript that one journal cannot offer to publish to another EMBO publication. The full manuscript and if applicable, reviewers reports are automatically sent to the receiving journal to allow for fast handling and a prompt decision on your manuscript. For more details of this service, and to transfer your manuscript to another EMBO title please click on Link Not Available

Referee #1 (Remarks for Author):

In the revised version of the manuscript Dallel S et al provide new information.

1. Regarding information on PCOS patients, the authors added details in Table S5, including available hormonal assays and ultrasound data, as well as information on patient recruitment at the Clermont-Ferrand hospital. To confirm the proposed model, the authors performed RT-qPCR on 9 targets and modified Figures 7G and S9C to include these results. They were unable to conduct immunocytochemical studies due to a lack of samples.

***Since the human samples were not available, we followed the editor's recommendations and performed investigations using a mouse model of PCOS. The results are now included in the article Figure 7 and Figure S12.

2. The authors explored granulosa cells in the context of PCOS using the KGN cell line and examined the accumulation of TXNIP and other inflammasome components. They observed heterogeneous results and did not find any regulation of TXNIP, leading them to conclude that they need to explore other human granulosa cell lines. It would be better to use primary culture of granulosa cells of PCOS patients.

***It is ethically and clinically not possible to conduct a study on primary human mural granulosa cells from PCOS patients since no ovarian sampling is performed on these patients.

3. Regarding the proposed role of MIR1224, the authors attempted to detect this microRNA by RT-qPCR but were unsuccessful, and thus decided to remove this part of the discussion. In conclusion the model proposed in rodent species is not confirmed in humans.

***This model has been proposed in the discussion (3 lines of text) and is not a crucial molecular mechanism on the paper is based. This absence of MIR1224 regulation downstream LXR just means that regulation of TXNIP depend on another pathway which needs to be identify.

We hope that the modifications we have made fully address the concerns raised and help lift any reservations regarding the acceptance of this article for publication in EMBO Molecular Medicine.

15th Apr 2025

Dear Dr. Baron,

Thank you for the submission of your manuscript to EMBO Molecular Medicine. We have now received feedback from the two reviewers who agreed to evaluate your manuscript. As you will see from their reports pasted below, while the referee #1 remains critical regarding the use of mural granulosa cells, the referee #3 supports publication of the manuscript. Therefore, I am pleased to inform you that we will be able to accept your manuscript pending the following final amendments:

1) Please address referee #1 concern by including the sentence in the main manuscript text acknowledging the limitation due to the lack of analysis in human mural granulosa cells.

2) Figures: Please consider assembling some of the supplementary figures to several expanded view figures. Please check our author guidelines for more information:

<https://www.embopress.org/page/journal/17574684/authorguide#expandedview>

3) Author checklist: Please submit a complete checklist. <https://www.embopress.org/pb-assets/embo-site/EMBO%20Press%20Author%20Checklist-1642513524327.xlsx>

4) In the main manuscript file, please do the following:

- Please address all comments suggested by our data editors listed below:

o Data availability statement:

1. Please note that the specific URLs for GSE22134, GSE222135, GSE271363 datasets are not provided in the data availability statement.

o Figure legends:

1. Please note that the exact p values are not provided in the legends of figures 1C, E, F; 2B, G, L; 4B, 5A, C, D; 6B, D, E.

2. Please indicate what ** represents; if this represents p value(s), please indicate the statistical test used and where appropriate, specify the exact p value in the legend(s) of figure(s) 3C.

3. Please indicate the statistical test used for data analysis in the legends of figures 4B, 7E, F.

4. Please note that the box plots need to be defined in terms of minima, maxima, centre, bounds of box and whiskers, and percentile in the legends of figures 1E, 3C, 5A, C, D, E; 6B.

5. Please note that information related to n is missing in the legends of figures 1C, E, F; 2B, G, L; 3C, 4E, 5A, C, D, E; 6B, E; 7E.

6. Please note that the error bars are not defined in the legends of figures 1C, F; 2B, G, L; 4E; 6E, 7E.

7. Please note that the scale bar needs to be defined for figure 3D.

8. Please note that the black arrows are not defined in the legend of figure 1B, D. This needs to be rectified.

9. Please note that the white arrows are not defined in the legend of figure 1D. This needs to be rectified.

- Add up to 5 keywords.

- Please make sure to call out all figures and tables incl. appendix figures and tables in sequential order.

- In Methods, provide the statement that informed consent was obtained from all human subjects and confirm that the experiments conformed to the principles set out in the WMA Declaration of Helsinki and the Department of Health and Human Services Belmont Report.

- In Methods, add statistical paragraph that should reflect all information that you have filled in the Authors Checklist, especially regarding randomization, blinding, replication etc.

- Rename Illustrations to Graphics.

- Indicate in legends exact n and exact p values, not a range, along with the statistical test used. To keep the figures "clear" some authors found providing an Appendix table Sx with all exact p-values preferable. You are welcome to do this if you want to.

- Please include structured Methods section that includes a Reagents and Tools Table (should be uploaded as a separate file) followed by a Methods and Protocols section. More information on how to adhere to this format as well as downloadable templates (.docx) for the Reagents and Tools Table can be found in our author guidelines:

<https://www.embopress.org/page/journal/17574684/authorguide#structuredmethods>

An example of a paper with Structured Methods can be found here:

<https://www.embopress.org/doi/full/10.1038/s44320-024-00037-6#sec-4>

- Rename "Competing interests" to "Disclosure Statement & Competing Interests". We updated our journal's competing interests policy in January 2022 and request authors to consider both actual and perceived competing interests. Please review the policy <https://www.embopress.org/competing-interests> and update your competing interests if necessary.

- Author contributions: Please remove it from the manuscript and specify author contributions in our submission system. CRediT has replaced the traditional author contributions section because it offers a systematic machine-readable author contributions format that allows for more effective research assessment. You are encouraged to use the free text boxes beneath each contributing author's name to add specific details on the author's contribution. More information is available in our guide to authors:

<https://www.embopress.org/page/journal/17574684/authorguide#authorshipguidelines>

- Please rename "Data and materials availability" to "Data Availability". Please use the following format to report the accession number of your data:

[data type]: [full name of the resource] [accession number/identifier] ([doi or URL or identifiers.org/DATABASE:ACCESSION])

Please check "Author Guidelines" for more information.

<https://www.embopress.org/page/journal/17574684/authorguide#availabilityofpublishedmaterial>

- Correct the reference citation in the text and reference list. In the text a reference should be cited by author and year of publication. Include a space between a word and the opening parenthesis of the reference that follows. In the reference list, citations should be listed in alphabetical order. Where there are more than 10 authors on a paper, 10 will be listed, followed by "et al.". Also, please remove DOIs. Please check "Author Guidelines" for more information.

<https://www.embopress.org/page/journal/17574684/authorguide#referencesformat>

5) Funding: Please make sure that information about all sources of funding are complete in both our submission system and in the manuscript.

6) Tables: Please rename Tables S1-S6 to Dataset EV1-EV6 and place their legends in a separate tab/worksheet.

7) Appendix: Please compile the rest of supplementary figures (see point 2) and Tables S7-S9 into Appendix with a table of contents, including page numbers on the title page. Rename them to Appendix Figure S1 etc. and Appendix Table S1 etc. and update their callouts in the main manuscript text.

8) Synopsis:

- Synopsis text: Please remove it from the main manuscript text and upload it as a separate .doc file.

- Synopsis image: Please provide the image as a high-resolution jpeg file 550 pixels wide x 200-600 pixels high.

9) Source data: Please upload source data that you submitted with your previous submission and please add source data for newly generated data (e.g. Figure 7I).

10) As part of the EMBO Publications transparent editorial process initiative (see our Editorial at

<http://embomolmed.embopress.org/content/2/9/329>), EMBO Molecular Medicine will publish online a Review Process File (RPF) to accompany accepted manuscripts. This file will be published in conjunction with your paper and will include the anonymous referee reports, your point-by-point response and all pertinent correspondence relating to the manuscript. Let us know whether you agree with the publication of the RPF and as here, if you want to remove or not any figures from it prior to publication. Please note that the Authors checklist will be published at the end of the RPF.

11) Please provide a point-by-point letter INCLUDING my comments as well as the reviewer's reports and your detailed responses (as Word file).

I look forward to reading a new revised version of your manuscript as soon as possible.

Zeljko Durdevic
Senior Editor
EMBO Molecular Medicine

*** Instructions to submit your revised manuscript ***

1) a .docx formatted version of the manuscript text (including Figure legends and tables)

2) Separate figure files*

3) supplemental information as Expanded View and/or Appendix. Please carefully check the authors guidelines for formatting Expanded view and Appendix figures and tables at <https://www.embopress.org/page/journal/17574684/authorguide#expandedview>

4) a letter INCLUDING the reviewer's reports and your detailed responses to their comments (as Word file).

5) The paper explained: EMBO Molecular Medicine articles are accompanied by a summary of the articles to emphasize the major findings in the paper and their medical implications for the non-specialist reader. Please provide a draft summary of your article highlighting

6) Author contributions: the contribution of every author must be detailed in a separate section.

7) EMBO Molecular Medicine now requires a complete author checklist (<https://www.embopress.org/page/journal/17574684/authorguide>) to be submitted with all revised manuscripts. Please use the checklist as guideline for the sort of information we need WITHIN the manuscript. The checklist should only be filled with page numbers where the information can be found. This is particularly important for animal reporting, antibody dilutions (missing) and exact values and n that should be indicated instead of a range.

8) Every published paper now includes a 'Synopsis' to further enhance discoverability. Synopses are displayed on the journal webpage and are freely accessible to all readers. They include a short stand first (maximum of 300 characters, including space) as well as 2-5 one sentence bullet points that summarise the paper. Please write the bullet points to summarise the key NEW findings. They should be designed to be complementary to the abstract - i.e. not repeat the same text. We encourage inclusion of key acronyms and quantitative information (maximum of 30 words / bullet point). Please use the passive voice. Please attach these in a separate file or send them by email, we will incorporate them accordingly.

You are also welcome to suggest a striking image or visual abstract to illustrate your article. If you do please provide a jpeg file 550 px-wide x 300-600px high.

9) A Conflict of Interest statement should be provided in the main text

10) Please note that we now mandate that all corresponding authors list an ORCID digital identifier. This takes <90 seconds to complete. We encourage all authors to supply an ORCID identifier, which will be linked to their name for unambiguous name identification.

Currently, our records indicate that the ORCID for your account is 0000-0002-4524-3087.

Please click the link below to modify this ORCID:
Link Not Available

11) Include a Reagents and Tools Table as part of the Methods section, which can be downloaded from our author guidelines (<https://www.embopress.org/page/journal/17574684/authorguide#structuredmethods>)

- Graphs 800-1,200 DPI
- Photos 400-800 DPI
- Colour (only CMYK) 300-400 DPI"

*Additional important information regarding figures and illustrations can be found at <https://bit.ly/EMBOPressFigurePreparationGuideline>. See also figure legend preparation guidelines: <https://www.embopress.org/page/journal/17574684/authorguide#figureformat>

***** Reviewer's comments *****

Referee #1 (Remarks for Author):

Authors can find several articles using primary cultures of mural granulosa cells here. Mural granulosa cells were obtained from follicular fluid aspirated during an IVF protocol. A specific request to an ethics committee could be made.

Differential Gene Expression Analysis of Human Ovarian Follicular Cumulus and Mural Granulosa Cells Under the Influence of Insulin in IVF Ovulatory Women and Polycystic Ovary Syndrome Patients Through Network Analysis. Pankaj Pant et al, Endocr Res . 2024

The in-vitro effect of gonadotropins' type and combination on Granulosa cells gene expressions Yuval Yung et al, Reprod Biol Endocrinol. 2022

The Differential Metabolomes in Cumulus and Mural Granulosa Cells from Human Preovulatory Follicles Er-Meng Gao et al, Reproductive Sciences 2022

Dysregulation of anti-Mullerian hormone expression levels in mural granulosa cells of FMR1 premutation carriers. Moran Friedman-Gohas et al, Scientific Reports 2021

The authors had previously observed that LXR DKO mice exhibit an OHSS-like phenotype in response to ovarian stimulation suggesting a protective role of LXR against hyperstimulation. The novelty for preclinical studies would be the confirmation of the model in humans.

Referee #3 (Comments on Novelty/Model System for Author):

I have no further comments for improvement.

Referee #3 (Remarks for Author):

I have no further comments.

***** Reviewer's comments *****

Referee #1 (Remarks for Author):

Authors can find several articles using primary cultures of mural granulosa cells here. Mural granulosa cells were obtained from follicular fluid aspirated during an IVF protocol. A specific request to an ethics committee could be made.

Differential Gene Expression Analysis of Human Ovarian Follicular Cumulus and Mural Granulosa Cells Under the Influence of Insulin in IVF Ovulatory Women and Polycystic Ovary Syndrome Patients Through Network Analysis. Pankaj Pant et al, Endocr Res . 2024

The in-vitro effect of gonadotropins' type and combination on Granulosa cells gene expressions Yuval Yung et al, Reprod Biol Endocrinol. 2022

The Differential Metabolomes in Cumulus and Mural Granulosa Cells from Human Preovulatory Follicles Er-Meng Gao et al, Reproductive Sciences 2022

Dysregulation of anti-Mullerian hormone expression levels in mural granulosa cells of FMR1 premutation carriers. Moran Friedman-Gohas et al, Scientific Reports 2021

The authors had previously observed that LXR DKO mice exhibit an OHSS-like phenotype in response to ovarian stimulation suggesting a protective role of LXR against hyperstimulation. The novelty for preclinical studies would be the confirmation of the model in humans.

***We thank Reviewer#1 for these relevant informations and will take them into account in our future clinical investigations. As suggested by the editor, the limitation of the human cohort exploration has been mentioned in the discussion section.

Referee #3 (Comments on Novelty/Model System for Author):

I have no further comments for improvement.

Referee #3 (Remarks for Author):

I have no further comments.

***We warmly thank Reviewer#3 for evaluating this work.

29th Apr 2025

Dear Dr. Baron,

We are pleased to inform you that your manuscript is accepted for publication and is now being sent to our publisher to be included in the next available issue of EMBO Molecular Medicine.

Zeljko Durdevic
Senior Editor
EMBO Molecular Medicine
